# Novel 3′-Substituted-1′,2′,4′-Oxadiazole Derivatives of 18βH-Glycyrrhetinic Acid and Their *O*-Acylated Amidoximes: Synthesis and Evaluation of Antitumor and Anti-Inflammatory Potential In Vitro and In Vivo

**DOI:** 10.3390/ijms21103511

**Published:** 2020-05-15

**Authors:** Andrey V. Markov, Aleksandra V. Sen’kova, Irina I. Popadyuk, Oksana V. Salomatina, Evgeniya B. Logashenko, Nina I. Komarova, Anna A. Ilyina, Nariman F. Salakhutdinov, Marina A. Zenkova

**Affiliations:** 1Institute of Chemical Biology and Fundamental Medicine, Siberian Branch of the Russian Academy of Sciences, Lavrent’ev ave., 8, 630090 Novosibirsk, Russia; alsenko@mail.ru (A.V.S.); ana@nioch.nsc.ru (O.V.S.); evg_log@niboch.nsc.ru (E.B.L.); humanity2206@mail.ru (A.A.I.); marzen@niboch.nsc.ru (M.A.Z.); 2N.N. Vorozhtsov Novosibirsk Institute of Organic Chemistry, Siberian Branch of the Russian Academy of Sciences, Lavrent’ev ave., 9, 630090 Novosibirsk, Russia; popadyuk@nioch.nsc.ru (I.I.P.); komar@nioch.nsc.ru (N.I.K.); anvar@nioch.nsc.ru (N.F.S.)

**Keywords:** 18βH-glycyrrhetinic acid, derivatives, oxadiazole, heterocyclic moiety, antitumor activity, anti-inflammatory activity, apoptosis, metastasis, target prediction, molecular docking

## Abstract

A series of novel 18βH-glycyrrhetinic acid (GA) derivatives containing 3′-(alkyl/phenyl/pyridin(-2″, -3″, and -4″)-yl)-1′,2′,4′-oxadiazole moieties at the C-30 position were synthesized by condensation of triterpenoid’s carboxyl group with corresponding amidoximes and further cyclization. Screening of the cytotoxicity of novel GA derivatives on a panel of tumor cell lines showed that the 3-acetoxy triterpenoid intermediates—*O*-acylated amidoxime **3a-h**—display better solubility under bioassay conditions and more pronounced cytotoxicity compared to their 1′,2′,4′-oxadiazole analogs **4f-h** (median IC_50_ = 7.0 and 49.7 µM, respectively). Subsequent replacement of the 3-acetoxy group by the hydroxyl group of pyridin(-2″, 3″, and -4″)-yl-1′,2′,4′-oxadiazole-bearing GA derivatives produced compounds **5f-h**, showing the most pronounced selective toxicity toward tumor cells (median selectivity index (SI) > 12.1). Further detailed analysis of the antitumor activity of hit derivative **5f** revealed its marked proapoptotic activity and inhibitory effects on clonogenicity and motility of HeLa cervical carcinoma cells in vitro, and the metastatic growth of B16 melanoma in vivo. Additionally, the comprehensive in silico study revealed intermediate **3d**, bearing the *tert*-butyl moiety in *O*-acylated amidoxime, as a potent anti-inflammatory candidate, which was able to effectively inhibit inflammatory response induced by IFNγ in macrophages in vitro and carrageenan in murine models in vivo, probably by primary interactions with active sites of MMP9, neutrophil elastase, and thrombin. Taken together, our findings provide a basis for a better understanding of the structure–activity relationship of 1′,2′,4′-oxadiazole-containing triterpenoids and reveal two hit molecules with pronounced antitumor (**5f**) and anti-inflammatory (**3d**) activities.

## 1. Introduction

Oxadiazoles—five-membered ring heterocycles containing two carbon atoms, two nitrogen atoms, and one oxygen atom—are significant structural elements of organic compounds that are interesting in the fields of industrial materials [1,2,3,4] and biologically active substances [4,5]. Modification of various chemical compounds with oxadiazole rings has several different purposes. On the one hand, these moieties act as planar aromatic linkers, placing substituents in the appropriate orientation. On the other hand, oxadiazole groups are important structural elements responsible for the emergence of new valuable properties in the obtained compounds.

Oxadiazole derivatives, especially 1,2,4- and 1,3,4-regiostereoisomers, are widely represented in medical chemistry. These compounds have been shown to possess anti-inflammatory [6,7], antidiabetic [8], antibacterial [9], antituberculosis [10], and anticancer [11,12,13] activities, among others.

These modifications can be used as replacements for carboxylate derivatives, such as esters, amides, carbamates, and hydroxamic esters, due to the stability of oxadiazolic rings in the aqueous solutions, which is one of the most important characteristics explaining the high interest in the use of oxadiazole moieties in the design of novel bioactive molecules. Moreover, the presence of nonligand electron pairs from heteroatoms in the structure of oxadiazoles allows them to act as acceptors of hydrogen bonds. Therefore, this bioisosteric substitution can improve the binding mode of oxadiazole-bearing derivatives with their biologically relevant protein targets [5].

Recently, considerable interest has been devoted to oxadiazole derivatives based on natural metabolites, including resveratrol [14], coumarins [15,16], monoterpenoids [17], steroids [18,19], and others (Figure 1, compounds **A**,**B**,**D**,**E**). In a large number of works, it was shown that introduction of various oxadiazole-substituted moieties significantly increased the biological activity of the parent molecules [14,15,16,17,18,19]. Additionally, some intermediate derivatives containing *O*-acylated amidoxime precursors are also found to display pronounced bioactive effects compared to starting compounds [17,19] (Figure 1, compounds **C**,**F**).

Interestingly, oxadiazole derivatives of pentacyclic triterpenoids have been investigated in a limited number of works [20,21,22,23,24,25], despite the fact that these extensively investigated natural metabolites are characterized by low systemic toxicity, ready availability, and multitargeting biological activities (Figure 1, compounds **G**-**J**). Moreover, biological evaluation of oxadiazole-bearing triterpenoids and their *O*-acylated amidoxime-containing intermediates was mainly limited to testing their cytotoxicity with respect to tumor cells, without identification of the probable mechanisms of action and animal studies. Gu et al. and Jin et al. identified some mechanisms underlying the cytotoxic effects of the most active derivatives of quinoline-fused ursolic acid; however, validation of their antitumor activity in vivo has not been shown [23,24].

In the present work, we report the design, synthesis, and study of the spectrum of biological activities of novel 18βH-glycyrrhetinic acid (GA) derivatives containing alkyl/phenyl/pyridinyl-1,2,4-oxadiazole rings instead of the native carboxylic group. Due to the fact that GA is characterized by a wide spectrum of pharmacological activities, with a predominance of those that are antitumor and anti-inflammatory [26], our attention was focused on the evaluation of the inhibitory activity of novel 1,2,4-oxadiazole-bearing GA derivatives and their *O*-acylated amidoxime-containing intermediates on tumor cell growth and inflammation—two processes that are highly interconnected with each other [27]. Here, we reveal that the introduction of pyridine substituents via the 1,2,4-oxadiazole group at the C-30 position of GA significantly improved the cytotoxic selectivity of novel derivatives for tumor cells. It is shown that the hit 3ʹ-pyridin-2ʹʹ-yl-1ʹ,2ʹ,4ʹ-oxadiazol of GA **5f** displays a complex effect on HeLa cervical carcinoma cells, including the induction of mitochondrial apoptosis, the inhibition of their clonogenic activity, and motility. Moreover, **5f** effectively inhibited metastatic growth of B16 murine melanoma in vivo in mice, without toxic effects. Additionally, the triterpenoid derivative **3d,** containing the *tert*-butyl moiety in the *O*-acylated amidoxime, was identified as a promising anti-inflammatory candidate. We show that **3d** effectively inhibits inflammatory response in IFNγ-stimulated macrophages in vitro and carrageenan-inflamed mice in vivo, probably by primary interactions with active sites of MMP9, neutrophil elastase, and thrombin.

## 2. Results

### 2.1. Synthesis of Novel Glycyrrhetinic Acid (GA) Derivatives

In order to improve the antitumor and anti-inflammatory properties of the parent GA molecule, we transformed the native carboxyl group of GA into bioisosteric 1,2,4-oxadiaxole with various aliphatic or aromatic substituents. The starting compound for the synthesis was 3-acetoxy-18βH-glycyrrhetinic acid **1** (Scheme 1). Given two well-known approaches to the synthesis of 1,2,4-oxadiazoles, including (a) the interaction of amidoximes with carboxylic acids or their derivatives and (b) the interaction of nitriles with nitrile oxides (1,3-dipolar addition) [28,29], we used the first method due to the availability of the starting reagents. The necessary amidoximes **2a-h**, containing different aliphatic or aromatic moieties, were synthesized according to literature data [30,31] from the corresponding nitriles and hydroxylamine.

Preliminary activation of the carboxyl group is necessary to carry out the reactions between compound **1** and amidoximes. Among a number of activating agents, *N,N*′-carbonyldiimidazole (CDI) was chosen, since it does not complicate further isolation and purification of target compounds. Activation of the carboxyl group of compound **1** with CDI and the subsequent reaction with *N*′-hydroxy(alkyl/aryl/hetaryl)imidamides **2a-h** led to the formation of the corresponding intermediates **3a-h** in quantitative yield. The ^13^C NMR spectra of compounds **3a-h** indicate the shift of the C-30 signal: the C-30 signal for **1** is located at 180 ppm. For products **3a-h**, *δ_C_* ~ 172.3–172.7 ppm. The ^1^H NMR spectra show the disappearance of the N–OH group signals of reagents **2a-h** and the COOH group signal of compound **1**.

The 1′,2′,4′-oxadiazole derivatives **4a-h** were obtained by cyclization of intermediates **3a-h** with tetrabutylammonium fluoride (TBAF), an effective catalyst for the preparation of 1,2,4-oxadiazoles [32] under reflux in tetrahydrofuran (THF), with yields from 57% to 76% after purification by flash column chromatography. According to the ^1^H NMR spectra, the formation of a 1,2,4-oxadiazole ring is evidenced by the disappearance of NH_2_ group signals of compounds **3a-h**. The ^13^C NMR spectra show a shift to a weak field of C-5′ signals by 10–12 ppm, C-3′ by 11–14 ppm, and C-29 by 1.5–2 ppm, and a shift to a higher field of the C-20 signal by 5.5–6 ppm. Subsequent alkaline hydrolysis of the 3-acetoxy group led to the formation of compounds **5a-h**, with 48–82% yields after purification.

The reagents used in the proposed synthesis are cheap and available; all reactions predominantly proceed with the formation of one product and are easily scalable. The structures of the new compounds were confirmed by ^1^H, ^13^C NMR, and high-resolution mass spectrometry.

### 2.2. Cytotoxicity of Novel GA Derivatives

On the first step of the biological evaluation of novel GA derivatives **3a-h, 4a-h**, and **5a-h**, we analyzed their cytotoxicity in a panel of cultured mammalian cells, including human cervical carcinoma HeLa and KB-3-1, human duodenal carcinoma HuTu-80, human lung adenocarcinoma A549, murine melanoma B16 cell lines, and nontransformed human fibroblasts hFF3. The cells were treated with derivatives for 48 h and cell viability was evaluated by 3-(4, 5-dimethylthiazol-2-yl)-2, 5-diphenyltetrazolium bromide (MTT) assay. Since the studied compounds contain two types of functional groups at position 3-acetoxy- or the hydroxyl-group, 3-acetoxy-GA **1** and **GA-Me** were used as references. The obtained IC_50_ values of compounds are summarized in Table 1. Additionally, hierarchical clustering of cytotoxic data was carried out in order to reveal groups of compounds with similar cytotoxic profiles (Figure 2A).

Firstly, in order to evaluate the effects on cytotoxicity of the conversion of the C-30 carboxylic group to the *O*-acylated amidoxome moieties bearing different substituents, the antiproliferative activities of compounds **3a-h** were compared with that of the starting substance **1**. As is evident from Table 1, the formation of *O*-acylated amidoxime with aliphatic substituents increased the cytotoxicity of derivatives in comparison to **1**. The mean IC_50_ values of compounds **3a-c,** bearing methyl, ethyl, and isopropyl groups, were found to decrease for hFF3, A549, and HeLa cells by 6, 4.8, and 4.3 times, respectively, compared to the IC_50_ of compound **1** (Table 1). HuTu-80 and KB-3-1 cells were less sensitive to this modification—the cytotoxic effect of compounds **3a-c** in these cell lines was only 1.3 times higher compared to compound **1**. Interestingly, the substitution of these moieties with bulky *tert*-butyl completely abolished the antiproliferative activity—the derivative **3d** displays IC_50_ values exceeding 50 µM with respect to all tested cell lines (Table 1, Figure 2A). Replacement of aliphatic moieties by aromatic groups decreased the cytotoxicity of compounds **3e-f** with respect to the tested malignant and nontransformed cells by an average of 2.3 and 3.6 times, respectively, in comparison with compounds **3a-c**. As can be seen in Table 1 and Figure 2A, the antiproliferative activity of aryl-bearing intermediates **3e-f** is virtually independent of the presence of the nitrogen atom and its position in the aromatic ring.

The formation of the 1,2,4-oxadiazole ring by cyclization of the *O*-acylated amidoximes significantly decreased the solubility of the compounds under bioassay conditions—among synthesized derivatives **4a-h**, only the pyridine-bearing derivatives **4f-h** were reasonably soluble, and therefore only their cytotoxicity was investigated. It was found that the formation of the oxadiazole ring decreased the cytotoxic activity—*O*-acylated amidoximes **3f-h** and their corresponding 1,2,4-oxadiazoles **4f-h** display median IC_50_ values of 7.4 µM and 49.7 µM, respectively, in all tested cell lines. Interestingly, the cytotoxicity of derivatives **4f-h** definitely depended on the position of the nitrogen atom in the heterocycle. In contrast to intermediates **3f-h** located in the same clade C in the obtained cladogram (Figure 1A), the corresponding 1,2,4-oxadiazole-bearing analogs **4f-h** display dramatically different cytotoxic profiles and are highly separated from each other in the phylogenic tree (Figure 1A, clades A, B, C). The median cytotoxicity of compounds **4f-h** decreased with the change of pyridin-2″-yl to pyridin-3″-yl and pyridin-4″-yl substituents. Among derivatives **4f-h**, compound **4f**, containing the pyridin-2ʹʹ-yl group, was the most toxic, with a median IC_50_ value of 11.5 µM, whereas the pyridin-3″-yl-substituted compound **4g** was markedly less active, with a median IC_50_ value of 49.9 µM (Table 1). The Introduction of the pyridin-4″-yl moiety was shown to completely abolish the antiproliferative activity; compound **4h** displays IC_50_ values equal to or exceeding 50 µM (Table 1, Figure 1A). Observed peculiarities in the cytotoxicity of **4f-h** can be explained by limitation of the conformational flexibility of pyridinyl substituents that probably affected the binding efficiency of the derivatives with their protein targets—hypothetically, the pyridin-2″-yl substituent contributed to forming a more favorable “protein–ligand” complex; however, additional investigations in this field are needed.

Deprotection of the hydroxy group at C-3 of 1,2,4-oxadiazole-bearing GA derivatives was found to increase their solubility; in addition to pyridine-containing compounds **5f-h**, *iso*-propyl-, *tert*-butyl-, and phenyl-bearing derivatives **5c-e** were also soluble. In comparison with **GA-Me**, novel 3-hydroxy GA derivatives **5c-h** were significantly less toxic; compound **5c**, being the most active in this group, displays a median IC_50_ value of 5.8 µM versus 1.1 µM for **GA-Me** (Table 1). Introduction of the bulky *tert*-butyl group significantly decreased cytotoxicity, as was shown above for *O*-acylated amidoximes **3c-d**; compound **5d** showed a median IC_50_ value of 16.5 µM. The replacement of aliphatic moieties with the phenyl group was undesirable for 3-hydroxy GA derivatives; compound **5e** exhibited no toxicity in all tested cell lines. Interestingly, the presence of a pyridine substituent instead of a phenyl one significantly changed the cytotoxic profiles of compounds; compounds **5f-h**, similarly to compound **5e**, were nontoxic to melanoma B16, lung carcinoma A549, and nontransformed hFF3 cells, but displayed marked cytotoxicity in cervical HeLa/KB-3-1 and duodenal HuTu-80 carcinoma cells (Table 1). Due to a significant difference between the sensitivity of nontransformed hFF3 and malignant HeLa, KB-3-1, and HuTu-80 cells to **5f-h**, these compounds were identified as the most selective among the investigated novel GA derivatives.

Indeed, calculation of the selectivity index (SI; the ratio between IC_50_ for the derivatives in nontransformed fibroblasts and IC_50_ in malignant cells) clearly showed that compounds **5f-h** are the three most selective derivatives (median SI**^5f-h^** > 12.1 in HeLa and HuTu-80 cells) (Figure 2B). Interestingly, **GA-Me** also displayed high SI values (median SI**^GA-Me^** = 10.9 in all tumor cells); however, its cytotoxicity in normal fibroblasts was significantly higher than the cytotoxicity of **5f-h** (Table 1), so **GA-Me** was not considered by us as a leader compound for further study.

Hierarchical clustering of the obtained cytotoxic data revealed three major clades of investigated derivatives (Figure 2A). Clade A includes nontoxic compounds **3d**, **4h**, and **5e**, clade B contains the highly selective derivatives **4g** and **5f-h**, and the most enriched clade (clade C) includes the remaining compounds, which are characterized by high or moderate nonspecific cytotoxicity (Figure 2A). Clustering of cell lines grouped them into two main clades and arranged the cells in line with the reduction of their susceptibility to investigated derivatives. The first clade includes the most sensitive HeLa, HuTu-80, and KB-3-1 cells displaying median IC_50_ values of 12.3, 13.4, and 15.5 µM, respectively. The second clade consists of B16, A549, and hFF3 cells, characterized by markedly lower sensitivity to the investigated compounds, with median IC_50_ values of 22.2, 24.9, and 23.9 µM, respectively.

As a result of the screening study, the 3’-pyridine-substituted 1’,2’,4’-oxadiazoles **5f-h** were identified as hit compounds characterized by the highest SI against cervical and duodenal carcinoma cells. Compound **5f,** containing the *o*-pyridine moiety, which displays a more pronounced antitumor effect, was chosen for the mechanistic studies on the most sensitive HeLa human cervical carcinoma cells.

### 2.3. Compound **5f** Triggers Mitochondrial Caspase-Dependent Apoptosis

On the next mechanistic block of the study, we questioned how the hit compound (**5f**) suppresses tumor cell viability. The two most well-characterized and prevalent methods of cell death, which are realized in controlled and uncontrolled regimens, are apoptosis and necrosis, respectively [33]. In the case of apoptosis, cells change their morphological characteristics, including cell rounding, decreased cell volume, chromatin condensation, nuclear fragmentation, and plasma membrane blebbing, which finally allows the reticuloendothelial system to phagocyte apoptotic cells with minimal impairment to the surrounding tissues. Unlike apoptosis, necrosis leads to the rupture of the cell membrane, with subsequent leakage of the cell contents into the extracellular area, resulting in inflammation and tissue damage [33]. Thus, the triggering of necrosis is highly unacceptable for compounds with antitumor activity. According to our recent review, pentacyclic triterpenoids can induce both the apoptosis and necrosis of tumor cells, with high prepotency in the programmed mode of cell death [34].

In order to assess the ability of compound **5f** to trigger apoptosis or necrosis in HeLa cells, we used double staining of cells by annexin V-fluorescein isothiocyanate (annexin V FITC) and propidium iodide (PI). As shown in Figure 3A, 1’,2’,4’-oxadiazole derivative **5f** dose-dependently induced the death of HeLa cells by apoptosis—incubation of tumor cells with compound **5f** at 7 µM for 48 h led to the accumulation of both early (quadrants Q4-4) and late (quadrants Q4-2) apoptotic cells to 12.5% and 60.7%, respectively, versus 3.0% and 3.2% in control cells. The increase of triterpenoid concentration to 10 µM caused a further transition of cells from the early to late phases of apoptosis (8.1% and 66.5% in Q4-4 and Q4-2, respectively); moreover, necrotic cells were detected (14.2%) (Figure 3A). The observed accumulation of necrotic cells in the group treated by **5f** at 10 µM is explained by its high dosage; previously, a similar dose-dependent conversion of apoptosis to necrosis was shown for both well-known anticancer agents (e.g., cisplatin, etoposide, and doxorubicin) [35,36,37] and a range of natural compounds [38,39,40]. Due to the domination of apoptosis in compound **5f**-induced cell death and a low number of necrotic cells that did not form a distinct necrotic population in the cytogram (Figure 3A), derivative **5f** can be considered as an antitumor hit compound.

In the next step of the study, we questioned whether mitochondrial dysfunction underlies compound-**5f**-induced cell death in HeLa cells. Analysis of mitochondrial membrane potential (∆ψ_m_) by ∆ψ_m_-sensitive JC-1 dye clearly showed that 1’,2’,4’-oxadiazole-containing triterpenoid **5f** triggers apoptosis through the mitochondrial-dependent pathway. As can be seen in Figure 3B, the compound dose-dependently increased the percentage of the cells with collapsed membrane potential—up to 72.5% and 85.2% in HeLa cells treated by **5f** at 7 and 10 µM, respectively, versus 9.5% in control cells.

The triggering of apoptosis is known to be highly interconnected with the activation of caspases, mediated the apoptotic signal transduction by cleavage of apoptosis-implicated proteins [41]. In order to assess the ability of compound **5f** to affect the caspase cascade, HeLa cells, treated with the investigated triterpenoid, were incubated with the fluorescein isothiocyanate (FITC) conjugate of the cell-permeable pancaspase inhibitor VAD-FMK (FITC-VAD-FMK), a fluorescent in situ marker for activated caspases. Flow cytometry analysis revealed that **5f** significantly increased the intensity of FITC fluorescence, which is associated with metacaspase activation in HeLa cells in a dose-dependent manner (Figure 3C). Additionally, the ability of **5f** to activate executioner caspases-3/7, which play key roles in the culminating steps of apoptosis, was revealed in an independent experiment (Figure 3D); using a conjugate of caspase-3/7 substrate tetrapeptide Asp-Glu-Val-Asp (DEVD) with aminoluciferin, we identified that the treatment of HeLa cells by compound **5f** at 5 and 7 µM caused a statistically reliable increase of caspase-3/7 activity by 1.3 and 1.9 times, respectively, in comparison with control (Figure 3D).

Thus, the performed mechanistic study clearly showed that the investigated 1’,2’,4’-oxadiazole containing triterpenoid **5f** induces cell death by activation of mitochondrial caspase-dependent apoptosis.

### 2.4. Compound **5f** Inhibits the Metastatic Potential of HeLa Cells In Vitro and the Metastatic Growth of Melanoma B16 In Vivo

In order to analyze the antitumor effect of compound **5f** more comprehensively, we assessed its ability to modulate the metastatic potential of tumor cells in vitro and in vivo. Firstly, we evaluated the influence of **5f** on the clonogenicity of HeLa cells, determining the capacity of single cells to grow into a colony that was associated with the stemness of tumor cells [42]. To evaluate this, HeLa cells were seeded at a low density with or without **5f** and incubated for 14 days, followed by colony visualization by microscopy after staining with crystal violet. As shown in Figure 4A, compound **5f** (0.5 µM) significantly (three-fold) suppressed the ability of HeLa cells to self-renew into colonies compared to the control. The 1′,2′,4′-oxadiazole containing triterpenoid **5f** used at a lower concentration had no effect on the clonogenic characteristics of HeLa cells (Figure 4A).

Based on the published data described the ability of GA and its derivatives to inhibit tumor cell motility, playing a crucial role in metastasis [43,44,45,46], we hypothesized that compound **5f** can modulate this process in HeLa cells. To test this, we evaluated the effect of the nontoxic concentration of **5f** on the migratory capacity of HeLa cells in real time mode. By measurement of the impedance on CIM-plate inserts, we found that **5f** statistically significant suppressed the motility of HeLa cells (Figure 4B).

Given the pronounced inhibitory effects of compound **5f** on metastasis-related processes revealed in vitro, we questioned whether the triterpenoid suppresses metastatic growth in animal models. In order to understand this, the antimetastatic activity of **5f** was studied on a metastatic model of murine B16 melanoma. Mice bearing B16 melanoma were intraperitoneally injected with **5f** three times a week, starting on day 4 after tumor cell inoculation. Five injections of the triterpenoid were carried out until sacrifice at 14 days after melanoma transplantation. Obtained results revealed that the administration of **5f** significantly decreased the numbers of surface metastases in the lungs of B16 melanoma-bearing mice by 3.4 and 4.5 times compared to vehicle-treated and untreated control animals, respectively (Figure 4C). Observed inhibition of metastasis development was also assessed by calculation of the metastasis inhibition index (MII). The MII of the control untreated group was taken as 0%, whereas the MII corresponding to 100% indicated the absence of metastases. The obtained MII value of the **5f**-treated group was found to be 77.7 ± 4.7% (Figure 4D). Surprisingly, the vehicle (10% Tween-80) also showed slight antimetastatic activity: the vehicle-treated group was characterized by an MII value of 28.4 ± 9.6% (Figure 4D). The revealed property of Tween-80 is consistent with published data; according to Schwartzberg and Navari, this surfactant can display modest antitumor activity [47]. Additionally, **5f** also blocked the development of the solitary surface metastases in other organs. In the control untreated group, liver and kidney metastases were found in 5/8 and 4/8 mice, respectively. The injection of the vehicle did not affect this process (5 out of 8 mice in this group had metastases in the liver and kidney), whereas in the group treated with **5f**, liver and kidney metastases were detected only in 1/9 and 2/9 mice, respectively. It should be noted that five-fold injection of **5f** at the used dosage did not cause toxic and unspecific immunomodulatory effects—the calculated organ indexes in all groups were similar (Figure 4E). Moreover, during the experiment, all **5f**-treated mice were alive, whereas 8/9 mice were alive in control untreated and vehicle-treated groups.

Thus, the obtained results proved that 3’-(pyridin-2″-yl)-1’,2’,4’-oxadiazole derivative **5f** displays a significant antitumor potential at different levels (cell death, inhibitory effects on tumor cell stemness and migration in vitro, and metastasis growth in vivo), which is evidence of the expediency of its further investigation.

### 2.5. Anti-Inflammatory Potential of Novel GA Derivatives

Given that GA and its derivatives display significant anti-inflammatory potential, along with their antitumor activity [48,49,50,51,52,53,54,55,56,57,58,59,60,61], we questioned in the second block of the study whether the novel GA derivatives—3’-substituted-1’,2’,4’-oxadiazoles and their *O*-acylated amidoximes—have anti-inflammatory properties. Firstly, in order to decline the number of processed compounds and only select hypothetically more bioactive molecules, an in silico approach was used. The structures of GA derivatives were uploaded to the PASS Online platform [62], where a spectrum of their bioactivities was predicted. Further, the probabilities of the anti-inflammatory activity of investigated derivatives, as well as their inhibitory effect on the production of proinflammatory mediator nitric oxide (II) (NO), were extracted from the obtained data. Subsequent ranking of the compounds by the PASS score revealed two leaders—substances **3a** and **3d**, displaying the highest anti-inflammatory potential (PASS score = 163.8 and 136.0 for anti-inflammatory and 134.7 and 136 for anti-NO activities, respectively) (Figure 5A). Due to the high cytotoxicity of compound **3a** (median IC_50_**^3a^** = 2.6 µM) (Table 1), only compound **3d**, being nontoxic in all tested cell lines (median IC_50_**^3d^** > 50 µM), was chosen for the subsequent validation of anti-inflammatory activity.

The performed MTT assay confirmed that compound **3d** did not affect cell viability—the treatment of murine RAW264.7 macrophages by **3d** at concentrations up to 50 µM for 24 h did not reveal cytotoxicity (Figure 5B).

In the next step, we evaluated whether use of compound **3d** at nontoxic concentrations inhibits the inflammatory response of macrophages to IFN-γ. To perform this, we treated IFN-γ-stimulated RAW264.7 cells with the derivative for 24 h, followed by measurement of proinflammatory mediators TNFα and NO in a culture medium using ELISA and Griess reaction assay, respectively. The obtained results confirmed the anti-inflammatory activity of **3d**; the treatment of activated macrophages by the triterpenoid dose-dependently suppressed the production of TNFα by 2.2–9.0 times compared to the IFN-γ-stimulated control (Figure 5C). Moreover, **3d** dose-dependently inhibited NO synthesis by IFN-γ-induced RAW264.7 cells by up to 5.3 times compared with the control (Figure 5D). Thus, compound **3d**, characterized by both low toxicity and significant anti-inflammatory activity in vitro, was selected as a leader for further evaluation in inflammation murine models.

### 2.6. Compound **3d** Effectively Inhibits Carrageenan-Induced Inflammation In Vivo

Given the anti-inflammatory activity of Compound **3d** observed in IFNγ-stimulated macrophages in vitro, we questioned whether this derivative inhibits inflammatory processes is induced in vivo. In order to understand this, the bioactivity of Compound **3d** was evaluated in carrageenan-inflamed mice.

Carrageenan-induced paw edema is one of the widely used experimental models of acute inflammation, based on the increase in local vascular permeability, hyperemic response, and cell migration under the influence of the well-known irritant carrageenan, which leads to localized swelling [63]. Acute inflammatory edema was induced by an subplantar injection of 1% carrageenan. Mice were pretreated with Compound **3d**, indomethacin, or a vehicle via gastric gavage 1 h before phlogogen administration. Analysis of edema growth 5 h after its induction revealed the significant anti-inflammatory activity of compound **3d**, which decreased the weight of edema by 1.7-fold compared with vehicle-treated mice (Figure 6A). The reference drug indomethacin was characterized by a similar efficacy, suppressing edema formation by three-fold in comparison with the control (Figure 6A).

Histological analysis of the footpads revealed that subplantar injection of carrageenan induced a local inflammatory reaction, including the development of a plethora of blood vessels, hemorrhages, and edema in the paw tissues, as well as their severe inflammatory infiltration, which was represented predominantly by neutrophils with an admixture of lymphocytes and macrophages (Figure 6B). The administration of compound **3d** was found to prevent carrageenan-induced inflammation, completely eliminating signs of circulatory disorders in the tissues (plethora and hemorrhages) and substantially reducing leukocyte infiltration in the dermis of the footpads, which was identical to the protection observed in indomethacin-treated mice (Figure 6B).

Further, the anti-inflammatory activity of compound **3d** was also independently validated in a murine model of carrageenan-induced peritonitis (Figure 6C,D). As illustrated in Figure 6C, an intraperitoneal injection of carrageenan caused acute peritonitis, characterized by a 2.5-fold increase in the total leukocyte number in the peritoneal exudate compared with control healthy mice. Interestingly, vehicle administration partially inhibited phlogogen-induced leukocyte migration into the peritoneal cavity; however, the level of leukocyte infiltration in the vehicle-treated group remained significantly higher than in untreated mice (Figure 6C). An analysis of leukocyte subpopulations in the peritoneal fluid revealed total neutrophil expansion in both carrageenan-injected untreated and vehicle-treated groups (Figure 6D). Intraperitoneal administration of **3d** was found to completely suppress inflammation in the peritoneal cavity and restore leukocyte subtype content to a healthy level (Figure 6D). The anti-inflammatory efficacy of the investigated derivative was similar to that of the reference drug dexamethasone—the treatment of inflamed mice by compound **3d** and dexamethasone was found to decrease the total leukocyte number in peritoneal exudates by 2.7- and 1.8-fold, respectively. However, unlike the triterpenoid, dexamethasone did not affect the distribution of the leukocyte subpopulation—leukocytes detected in the peritoneal fluid of dexamethasone-treated mice were mainly represented by neutrophils similarly to carrageenan-stimulated untreated and vehicle-treated groups (Figure 6D).

Thus, the obtained results clearly confirmed the anti-inflammatory activity of compound **3d**. Moreover, the investigated triterpenoid more effectively inhibited the development of carrageenan-induced acute peritoneal inflammation than the well-known steroid anti-inflammatory drug dexamethasone.

## 3. Discussion

Pentacyclic triterpenoids, characterized by a multitargeting mode of action and relatively low systemic toxicity, are considered as a promising source of novel pharmaceutical agents. Extensive work is currently underway to develop novel, highly bioactive compounds based on natural triterpenoid scaffolds [34]. The majority of published studies in this field have focused on evaluations of the antitumor potential of triterpenoid derivatives [34]. Additionally, the chemical derivatization of triterpenoid was also applied to develop compounds with pronounced anti-inflammatory, antiviral, antibacterial, and antidiabetic bioactivities, among others [64,65,66,67].

According to previous studies, the introduction of heterocyclic moieties to triterpenoids is a promising approach that could improve their biological effects [68]. For instance, it was shown that the addition of oxadiazole-containing moieties to triterpenoids markedly increased their cytotoxicity against tumor cells [24,25], enhanced selectivity for leukemia cells [69], and reinforced anti-inflammatory properties [70,71].

In this work, a series of 3′-substituted-1′,2′,4′-oxadiazole GA derivatives and their *O*-acylated amidoximes were synthesized and evaluated for inhibitory activity against two highly interconnected processes, including tumor cell growth and inflammation. Analysis of the cytotoxicity of novel compounds in a panel of tumor cell lines revealed that intermediates **3a-c** (*O*-acylated amidoximes) display the most pronounced cytotoxic profiles (median IC_50_**^3a-c^** = 3.0 µM) (Figure 2A). It should be noted that the replacement of alkyl substituents by aromatic groups (compounds **3f-h**) and the subsequent cyclization of the linker to 1,2,4-oxadiazole (compounds **4f-h**) markedly decreased the cytotoxicity of derivatives (median IC_50_**^3f-h^** = 7.4 µM, median IC_50_**^4f-h^** = 49.7 µM), along with their solubility (Figure 2A). The obtained results are in line with published data; similar SAR trends were detected by researchers from Nanjin Forestry University on 1,3,4-oxadiazole-bearing derivatives of quinoline-fused ursolic acid [23,24]. The detected decline in the cytotoxicity of pyridine-substituted 1,2,4-oxadiazole-containing compounds **4f-h** in comparison with the parent compound **1** was also consistent with published studies—previously, a comparable decrease of toxicity was observed after the addition of aromatic moieties via 1,3,4-oxadiazole linker to ursolic [22,70], quinoline-fused ursolic [23], and betulonic [20] acids.

The transformation of the 3-acetoxy group to the 3-hydroxyl group of pyridine-bearing GA derivatives produced the most selective compounds (**5f-h**) (median SI**^5f-h^** > 12.1 (HeLa, HuTu-80 cells)) among all tested semisynthetic triterpenoids. Based on the hierarchical clustering of cytotoxic data, we suppose that the pyridine group can mediate the observed reinforcement of tumor cell targeting of the compounds; only pyridine-containing derivatives **4g** and **5f-h** were shown to form the most tumor-selective clade (clade B) (Figure 2A). This is in accordance with published data; a similar increase of selectivity was previously detected after the introduction of the pyridine moiety to betulinic and 23-hydroxybetulinic acids via triazol, azine, or acetyl linkers [72,73,74].

Mechanistic studies of the hit compound (**5f**) showed its complex effect on tumor cells; compound **5f** at low dosage suppressed tumor cell clonogenicity and motility (Figure 4A,B), thus inhibiting the metastatic potential of malignant cells, whereas high concentrations of **5f** induced cell death by mitochondrial caspase-dependent apoptosis. The antitumor potential of compound **5f** revealed in vitro was further validated in a murine metastatic model of B16 melanoma (Figure 4C,D). Our data established that five injections of **5f** (50 mg/kg) statistically significantly suppressed metastasis growth by 70% compared to the vehicle-treated control (Figure 4D), without systemic toxicity (Figure 4E). The obtained data clearly indicate the high antitumor potential of **5f** and confirm the expediency of its more detailed investigation. As far as we are aware, this is the first reported validation of antitumor activity of 1,2,4-oxadiazole-bearing triterpenoids in a murine model in vivo.

Along with the antitumor potential, GA displayed significant anti-inflammatory activity in vitro in lipopolysaccharide (LPS)- and trychophytin-activated RAW264.7 macrophages [48,49,50,51] and fructose-stimulated HK2 kidney cells [52], as well as in a panel of murine models [52,53,54,55,56]. Some GA derivatives also showed good anti-inflammatory effects in LPS-challenged macrophages in vitro [57] and experimental inflammation induced by histamine, carrageenan, influenza A, or CCl_4_ in mice [58,59,60,61]. Thus, considering the marked anti-inflammatory potential of GA and its derivatives, we questioned in the second block of our study whether the compounds displaying a promising anti-inflammatory effect could be found among investigated molecules. An in silico bioactivity prediction study revealed that the derivative **3d**, bearing the bulky *tert*-butyl group in *O*-acylated amidoxime (*N*′-acyloxypivalimidamide moiety), can be considered as a probable anti-inflammatory candidate (Figure 5A). The performed experiments on IFNγ-stimulated murine RAW264.7 macrophages confirmed the obtained computational data—compound **3d** at nontoxic concentration effectively inhibited the production of proinflammatory TNFα and NO by inflamed RAW264.7 cells (Figure 5C,D). Moreover, the injections of **3d** effectively suppressed acute inflammation induced in mice by carrageenan, inhibiting phlogogen-induced local inflammatory reaction in the paws (Figure 6A,B) and neutrophil migration into the peritoneal cavity in the case of carrageenan-induced peritonitis (Figure 6C,D). As far as we know, our report demonstrates for the first time a strong anti-inflammatory effect of intermediates of oxadiazole-bearing triterpenoids.

In order to shed light on the possible mechanism of compound **3d** action, we attempted to reveal its probable primary protein targets. To understand these targets, the ligand-based virtual screening approach was used, based on the analysis of the structural similarity of the investigated compound with known drugs and inhibitors. According to a recent report by Awale and Reymond, the usage of Naïve–Bayes (NB) machine learning together with classical nearest neighbor (NN) search produced more valid results during target prediction in comparison with a standard technique based on NN analysis only [75]. Therefore, the combination of NB machine learning with NN approaches was applied to predict protein targets of compound **3d**, using the Polypharmacology Browser 2.0 tool [75]. The performed analysis revealed 20 probable targets of the *O*-acylated amidoxime **3d** listed in Table 2.

Annotation of identified proteins using GeneCards database (https://www.genecards.org/) showed their association with a wide spectrum of biological processes (Appendix A). Thus, we further questioned which proteins from the obtained list are the most related to the inflammatory response. To understand this, we reconstructed the protein–protein interaction (PPI) network based on revealed protein targets of **3d** and rodent inflammatome, a list of differentially expressed genes identified previously by Wang et al. in 11 independent rodent inflammatory disease models [76]. The PPI network, including 1705 nodes and 15,796 edges, was created based on the Search Tool for Retrieval of Interacting Genes/Proteins (STRING) database using Cytoscape (Figure 7). Next, we ranged probable targets of **3d** according to their level of interconnection (Figure 7) and detected five proteins (MMP9, ELANE, CNR2, F2, and IKBKB) in the center of the PPI network, which are characterized by a high degree, and therefore play an important role in murine inflammation.

To further validate the ability of **3d** to bind to the revealed inflammation-related key nodes, molecular docking simulations were performed. Our results showed that matrix metalloproteinase 9 (MMP9), neutrophil elastase (ELANE), and thrombin (F2) can be considered as probable primary targets of the investigated triterpenoid; compound **3d** can fit into the ligand binding pockets of the mentioned proteins with binding energy comparable with that of their known inhibitors (Figure 8A). The binding of compound **3d** to cannabinoid receptor 2 (CNR2) and IκB kinase β (IKBKB) was characterized by high values of binding energy, which clearly indicates there is no direct interaction.

As shown in Figure 8B, **3d** can dock well into the active site of MMP9; **3d** was found to occupy the S1′ substrate binding pocket by forming hydrophobic interactions with Leu187 and Pro421 [77]. Additionally, the triterpenoid core is stabilized by a range of hydrophobic residues, including the imidazolyl side chain of His401 and His411, participated in the catalytic zinc binding. Moreover, the N1 nitrogen and amino groups included in the side moiety of compound **3d** form two strong hydrogen bonds with the key catalytic components of MMP9—zinc ion (2.61 Å) and His401 (3.30 Å), responsible for zinc coordination. The bulk substituent of the investigated triterpenoid was also shown to hydrophobically interact with the catalytically essential Glu402 (Figure 8B). [77].

The top-scoring binding pose of compound **3d** in the neutrophil elastase (ELANE) is characterized by the triterpenoid scaffold residing in a hydrophobic pocket (Figure 8B), whereas the N1 nitrogen of the amidoxime moiety of **3d** forms a hydrogen bond with Ser195 (2.8 Å), being a key component of the catalytic triad of the enzyme [78]. Moreover, the carbonyl group of compound **3d** was positioned close to the catalytic site of ELANE, forming a hydrogen bond with Asn61 (2.99 Å).

In the case of thrombin (F2), compound **3d** was revealed to dock well into the active site of the protein, forming two hydrogen bonds with Gly216 (2.98 Å, 3.21 Å), which is an important residue for a series of inhibitors of hydrogen bonding to thrombin [79]. Moreover, the investigated triterpenoid forms hydrophobic interactions with residues His57 and Ser195, being the components of the catalytic triad; and Tyr60A, Pro60C, and Trp60D, contained in the substrate recognition 60-loop, which are important for thrombin recognition by small synthetic inhibitors [80]. Additionally, the triterpenoid core is stabilized by a range of hydrophobic residues, including Ser214 and Glu217, playing a crucial role in the interaction of peptide substrate with thrombin [81] (Figure 8B).

Thus, the obtained data showed that the anti-inflammatory activity of compound **3d** can be determined by its multi-target mode of action. Based on docking data, we found that the side chain of the investigated derivative plays a key role in its interaction with analyzed proteins; the *O*-acylated amidoxime linker forms hydrogen bonds with amino acid residues in active sites of the proteins, whereas the bulky *tert*-butyl group stabilized the positioning of the *O*-acylated amidoxime group by hydrophobic interactions (Figure 8B). The performed in silico analysis clearly showed that matrix metalloproteinase MMP9, neutrophil elastase, and thrombin—playing important roles in inflammatory processes [82,83,84]—can be considered as primary targets of **3d**; however, to elucidate this anti-inflammatory mechanism more precisely, a detailed biochemical or biophysical analysis is required.

The revealed ability of **3d** to be docked well to active sites of MMP9, neutrophil elastase, and thrombin correlated well with the observed inhibition of leukocyte migration into the peritoneum after treatment of carrageenan-inflamed mice by **3d** (Figure 6C,D). Previously, it was shown that MMP9 and neutrophil elastase play key roles in the regulation of inflammation-associated leukocyte chemotaxis by mediating interleukin-8 processing [82,85]. Moreover, MMP9 was found to participate in the formation of the chemotactic gradient in the extracellular matrix near the inflammatory site; this protease can cleave syndecan-1, which binds chemokines synthesized in response to inflammatory stimuli, and as a result stimulates their release, thereby providing directional cues to migrating neutrophils [82,86]. Neutrophil elastase can also affect the influx of inflammatory cells through the cleavage of laminin-332 and VEGF, with the production of fragments being chemotactic for neutrophils and macrophages, respectively [87,88]. Additionally, thrombin, one of the key participants in the coagulation cascade, can also be involved in the control of neutrophil migration. Previously, Tsen et al. showed that a range of thrombin-specific inhibitors significantly decreased the adhesion of neutrophils to endothelial cells and increased their rolling velocity [89], thereby diminishing the transmigration of neutrophils from blood vessels into inflamed sites. Thus, we supposed that the anti-inflammatory properties of compound **3d** could be due to its inhibitory effect on the activity of MMP9, neutrophil elastase, and thrombin, which can lead to the dysregulation of optimal chemokine processing and the generation of that chemotactic gradient that guides neutrophils into an inflamed site, as well as the suppression of transendothelial neutrophil migration.

The observed inhibitory effect of **3d** on the inflammatory response of RAW264.7 macrophages induced by IFNγ (Figure 5C,D) could be also associated with its probable inhibitory effect on neutrophil elastase. Previously, Hagiwara et al. revealed that the neutrophil elastase inhibitor sivelestat effectively inhibited the secretion of inflammatory mediators and NO by LPS-challenged RAW264.7 cells [90]; however, additional studies to test this hypothesis are required.

In conclusion, the performed cytotoxicity screening of novel 3′-substituted-1′,2′,4′-oxadiazole-containing GA derivatives and their intermediates revealed the hit compound (**5f**), bearing 3ʹ-pyridin-2″-yl substituent, which displays high antitumor selectivity and complex effects on HeLa cervical carcinoma cells, including the inhibition of their clonogenicity and motility, and a triggering of the intrinsic apoptotic pathway. The antitumor potential of **5f** was further validated in vivo; the investigated compound effectively inhibited the metastatic development of highly aggressive B16 melanoma in mice. Additionally, comprehensive in silico analysis of synthesized GA derivatives revealed intermediate **3d**—containing the *tert*-butyl group in the *O*-acylated amidoxime moiety—as a promising anti-inflammatory candidate able to bind to active sites of MMP9, neutrophil elastase, and thrombin. The performed experiments on cellular and murine models clearly confirmed the obtained computational data; it was found that **3d** effectively inhibited inflammatory response in RAW264.7 macrophages in vitro and carrageenan-induced inflammation in mice. Altogether, our findings provide new insights into the structure–activity relationship of heterocycle-containing triterpenoids and demonstrate that both oxadiazole-bearing compounds and their intermediates can display promising bioactivities.

## 4. Materials and Methods

### 4.1. General Experimental Procedures

Melting points were determined on a Mettler Toledo FP900 thermosystem and were uncorrected. The elemental composition of the products was determined from high-resolution mass spectra (HRMS) recorded on a double-focusing sector (DFS) Thermo Electron Corporation instrument. Optical rotations were measured with a PolAAr 3005 polarimeter. ^1^H and ^13^C NMR spectra were measured on Bruker spectrometers: DRX-500 (500.13 MHz for ^1^H and 125.76 MHz for ^13^C) and AV-400 (400.13 MHz for ^1^H and 100.61 MHz for ^13^C). Solutions of each compound were prepared in CDCl_3_. Chemical shifts were recorded in *δ* (ppm) using *δ* 7.24 (^1^H NMR) and *δ* 76.90 (^13^C NMR) of CHCl_3_ as internal standards. Chemical shift measurements were given in ppm and the coupling constants (*J*) were given in hertz (Hz). The structure of the compounds was determined by NMR using standard one-dimensional and two-dimensional procedures (^1^H-^1^H COSY, ^1^H-^13^C HMBC/HSQC, ^13^C-^1^H HETCOR/COLOC). The hydrogen or carbon atom assignments marked with the same *, ^#^, ^§^, ^‡^, or ^†^ symbols are interchangeable. The purity of the final compounds and intermediates for biological testing was >95%, as determined by HPLC analysis. HPLC analyses were carried out on a MilichromA-02 using a ProntoSIL 120-5-C18 AQ column (BISCHOFF, 2.0 × 75 mm column, grain size 5.0 lm). The mobile phase used Millipore purified water with 0.1% trifluoroacetic acid at a flow rate of 150 µL/min at 35 °C, with UV detection at 210, 220, 240, 260, and 280 nm. A typical run time was 25 min, with a linear gradient of 0–100% methanol. Flash column chromatography was performed with silica gel (Merck, 60–200 mesh). All courses of all reactions were monitored by TLC analysis using Merck 60 F254 silica gel on aluminum sheets with the eluents CHCl_3_ and CHCl_3_–MeOH (25:1.5).

### 4.2. Reagents

*N,N*′-carbonyldiimidazole (CDI), 2-Cyanopyridine, 3-cyanopyridine, 4-cyanopyridine, propionitrile, and trimethylacetonitrile were purchased from ACROS organics. Isobutyronitrile was purchased from Alfa Aesar and TBAF was purchased from Sigma-Aldrich. All solvents used in the reactions were purified and dried according to previously reported procedures. *N*′-hydroxy(alkyl/aryl/hetaryl)imidamides were prepared according to literature methods [30,31].

### 4.3. General Procedure A for 3β-acetoxy-11-oxo-18βH-olean-12-en-29-oic acid (**1**) Interaction with N′ Hydroxy(alkyl/aryl/hetaryl)Imidamides

The 3β-acetoxy-11-oxo-18βH-olean-12-en-29-oic acid (**1**) (1 equivalent (equiv.)) and CDI (1.2 equiv.) were dissolved in dry CH_2_Cl_2_ and stirred for 2.5 h. Then, corresponding N′-hydroxy(alkyl/aryl/hetaryl)imidamides (1.2 equiv.) was added to the solution and the reaction mixture was stirred until complete conversion was reached. The reaction course was monitored by TLC (CHCl_3_–MeOH, 25–0.5). The reaction mixture was evaporated until dry, redissolved in CHCl_3_, and chromatographed on silica gel.

### 4.4. General Procedure B for Cyclization of Compounds **3a–3h** by n-Bu4NF in THF

Corresponding *N*′-(3β-acetoxy-11-oxo-18βH-olean-12-en-29-oyl)(alkyl/aryl/hetaryl)imidamide **3a-h** (1 equiv.) was dissolved in THF, tetrabutylammonium fluoride 1M solution in THF (1 equiv.) was added, and the reaction mixture was refluxed until complete conversion was reached. The reaction course was monitored by TLC (CHCl_3_–MeOH, 25–0.5). The reaction mixture was evaporated until dry and redissolved in CHCl_3_–AcOEt. The organic layer was washed with brine, dried over anhydrous MgSO_4_, and evaporated until dry, obtaining a crude product.

### 4.5. General Procedure C for the Hydrolysis of Acetate Groups of Oxadiazole Derivatives **4a–4h**

The mixture of corresponding 30-nor-3β-acetoxy-11-oxo-20-(3′-alkyl/aryl/heteroaryl-1′,2′,4′-oxadiazol-5′-yl)-18βH-olean-12-en **4a-h** (1 equiv.) and KOH (6 equiv.) in MeOH was refluxed until complete conversion was reached. The reaction course was monitored by TLC (CHCl_3_–MeOH, 25–0.5). Then, the reaction mixture was cooled to room temperature and HCl (5% aq. solution) was added drop-wise to pH 7. Methanol was evaporated, then the crude product was dissolved in CH_2_Cl_2_–AcOEt, washed with saturated aqueous NaHCO_3_, brined, and dried over anhydrous MgSO_4_. Solvent was removed to give a crude product as a yellowish amorphous solid.

### 4.6. N′-(3β-Acetoxy-11-oxo-18βH-olean-12-en-29-oyl)Acetimidamide (**3a**)

Product **3a** (2.2 g, quantitative yield) was obtained as a white amorphous solid according to the general procedure **A** from compound **1** (2.0 g, 3.9 mmol), dry CH_2_Cl_2_ (20 mL), CDI (0.76 g, 4.69 mmol), and *N*′-hydroxyacetimidamide (0.35 g, 4.69 mmol). Crude product **3a** was purified by flash column chromatography (SiO_2_, 0–4% MeOH in CHCl_3_). Mp 112.8–117.3 °C. [$\alpha_{D}^{25}$] +94 (*c* 0.20 g/100 mL; CHCl_3_). high-resolution mass spectra (HRMS): m/z calc. for (C_34_H_52_O_5_N_2_)^+^ 568.3871; found 568.3876. ^1^H NMR (CDCl_3_, 400 MHz): *δ* = 5.54 (s, 1H, H-12), 4.99 (br.s, 2H, NH_2_), 4.42 (dd, 1H, *J_3a,2a_* = 11.6, *J_3a,2e_* = 4.7, H-3a), 2.69 (dm, 1H, *J_1e,1a_* = 13.4, H-1e), 2.27 (s, 1H, H-9), 2.12 (m, 1H, H-18), 2.03-1.90 (m, 8H; H-21, H-15a, 1.97 (s, 3H, CH_3_-32), 1.92 (s, 3H, CH_3_-1′′)), 1.87 (m, 1H, H-19), 1.75 (m, 1H, H-16a), 1.69-1.45 (m, 5H; H-19′, H-2, H-7, H-6, H-2′), 1.45-1.23 (m, 8H; H-22, H-22′, H-7′, H-21′, H-6′, 1.29 (s, 3H, CH_3_-27)), 1.17 (s, 3H, CH_3_-29), 1.11 (dm, 1H, H-16e), 1.07 (s, 3H, CH_3_-25), 1.04 (s, 3H, CH_3_-26), 1.01-0.90 (m, 2H; H-1a, H-15e), 0.80 (s, 6H, CH_3_-23, CH_3_-24), 0.73 (s, 3H, CH_3_-28), 0.73 (m, 1H, H-5a). ^13^C NMR (CDCl_3_, 100 MHz): *δ* = 199.58 (s, C-11), 172.68 (s, C-30), 170.76 (s, C-31), 169.31 (s, C-13), 155.49 (s, C-3′), 128.02 (d, C-12), 80.33 (d, C-3), 61.45 (d, C-9), 54.70 (d, C-5), 48.07 (d, C-18), 45.10 (s, C-14), 43.84 (s, C-20), 42.93 (s, C-8), 41.23 (t, C-19), 38.49 (t, C-1), 37.73 (s, C-4), 37.24 (t, C-22), 36.64 (s, C-10), 32.38 (t, C-7), 31.64 (s, C-17), 31.11 (t, C-21), 28.44 (q, C-28*), 28.00 (q, C-29*), 27.76 (q, C-23), 26.18 (t, C-16^§^), 26.09 (t, C-15^§^), 23.26 (t, C-2), 23.03 (q, C-27), 21.06 (q, C-32), 18.40 (q, C-26), 17.07 (t, C-6), 16.77 (q, C-1′′), 16.40 (q, C-24), 16.11 (q, C-25).

### 4.7. 30-nor-3β-acetoxy-11-oxo-20-(3′-methyl-1′,2′,4′-oxadiazol-5′-yl)-18βH-olean-12-en (**4a**)

Crude product **4a** (1.75 g, 86%) was obtained according to the general procedure **B** from compound **3a** (2.1 g, 3.70 mmol), Bu_4_NF 1M solution in THF (3.7 mL, 3.70 mmol), and THF (20 mL). Crude product **4a** was purified by flash column chromatography (SiO_2_, 0–2% MeOH in CHCl_3_), recrystallized from EtOH, and then from AcOEt to give a pure sample of **4a** (1.2 g, 59%). Mp 313.4–315.0 °C (AcOEt). [$\alpha_{D}^{23.5}$] +183 (*c* 0.20 g/100 mL; CHCl_3_). HRMS: m/z calc. for (C_34_H_50_O_4_N_2_)^+^ 550.3765; found 550.3764. ^1^H NMR (CDCl_3_, 400 MHz): *δ* = 5.68 (s, 1H, H-12), 4.48 (dd, 1H, *J_3a,2a_* = 11.5, *J_3a,2e_* = 4.7, H-3a), 2.76 (dm, 1H, *J_1e,1a_* = 13.6, H-1e), 2.36 (s, 3H, CH_3_-1′′), 2.33 (s, 1H, H-9), 2.21-1.95 (m, 7H; H-21, H-18, H-19, H-15, 2.02 (s, 3H, CH_3_-32)), 1.85 (ddd, 1H, J = J = 14.3, H-19′), 1.80 (m, 1H, H-16), 1.71-1.49 (m, 5H; H-2, H-7, H-2′, H-21, H-6), 1.48-1.31 (m, 6H; H-6′, H-22, H-7′, 1.36 (s, 3H, CH_3_-27)), 1.30-0.95 (m, 13H; 1.26 (s, 3H, CH_3_-29), H-22′, H-16′, 1.12 (s, 3H, CH_3_-25), 1.08 (s, 3H, CH_3_-26), H-1′a, H-15′), 0.85 (s, 6H, CH_3_-23, CH_3_-24), 0.77 (dm, 1H, H-5a), 0.72 (s, 3H, CH_3_-28)). ^13^C NMR (CDCl_3_, 100 MHz): *δ* = 199.78 (s, C-11), 183.47 (s, C-5′(30)), 170.87 (s, C-31), 168.30 (s, C-13), 166.96 (s, C-3′), 128.62 (d, C-12), 80.44 (d, C-3), 61.58 (d, C-9), 54.86 (d, C-5), 47.34 (d, C-18), 45.28 (s, C-14), 43.00 (s, C-8), 41.39 (t, C-19), 38.62 (t, C-1), 38.34 (s, C-20), 37.90 (s, C-4), 36.95 (s, C-10), 36.76 (t, C-22), 32.53 (t, C-7), 31.75 (t, C-21), 31.69 (s, C-17), 30.00 (q, C-29), 28.14 (q, C-28), 27.90 (q, C-23), 26.25 (t, C-16*), 26.25 (t, C-15*), 23.41 (t, C-2), 23.31 (q, C-27), 21.19 (q, C-32), 18.50 (q, C-26), 17.22 (t, C-6), 16.54 (q, C-24), 16.27 (q, C-25), 11.57 (q, C-1″).

### 4.8. 30-nor-3β-hydroxy-11-oxo-20-(3′-methyl-1′,2′,4′-oxadiazol-5′-yl)-18βH-olean-12-en (**5a**)

Crude product **5a** (0.481 g, quantitative yield) was obtained according to the general procedure **C** from compound **4a** (0.5 g, 0.91 mmol), KOH (0.30 g, 5.45 mmol), and MeOH (25 mL). Crude product **3a** was purified by flash column chromatography (SiO_2_, 0–1% MeOH in CHCl_3_) and recrystallized from MeOH–CHCl_3_ to give a pure sample of **5a** (0.38 g, 82%). Mp 300.2–300.4 °C. [$\alpha_{D}^{20}$] +181 (*c* 0.20 g/100 mL; CHCl_3_). HRMS: m/z calc. for (C_32_H_48_O_3_N_2_)^+^ 508.3660; found 508.3651. ^1^H NMR (CDCl_3_, 400 MHz): *δ* = 5.68 (s, 1H, H-12), 3.20 (dd, 1H, *J_3a,2a_* = 10.2, *J_3a,2e_* = 5.7, H-3a), 2.76 (dm, 1H, *J_1e,1a_* = 13.4, H-1e), 2.36 (s, 3H, CH_3_-1″), 2.32 (s, 1H, H-9), 2.21-1.97 (m, 4H; H-21, H-18, H-19, H-15), 1.92-1.74 (m, 2H; H-19′, H-16), 1.74-1.50 (m, 5H; H-7, H-2, H-2′, H-6, H-21′), 1.50-1.32 (m, 6H; H-22, H-6′, H-7′, 1.37 (s, 3H, CH_3_-27)), 1.30-1.14 (m, 5H; 1.26 (s, 3H, CH_3_-29), H-22′, H-16′), 1.10 (s, 3H, CH_3_-25), 1.08 (s, 3H, CH_3_-26), 1.05-0.90 (m, 5H; H-15′, 0.97 (s, 3H, CH_3_-23), H-1′), 0.77 (s, 3H, CH_3_-24), 0.73 (s, 3H, CH_3_-28), 0.67 (dm, 1H, H-5). ^13^C NMR (CDCl_3_, 100 MHz): *δ* = 199.96 (s, C-11), 183.49 (s, C-5′(30)), 168.31 (s, C-13), 166.97 (s, C-3′), 128.65 (d, C-12), 78.61 (d, C-3), 61.68 (d, C-9), 54.77 (d, C-5), 47.33 (d, C-18), 45.27 (s, C-14), 43.01 (s, C-8), 41.42 (t, C-19), 38.99 (t, C-1*), 38.96 (s, C-4*), 38.35 (s, C-20), 36.95 (t, C-22^#^), 36.91 (s, C-10^#^), 32.58 (t, C-7), 31.74 (s, C-17^§^), 31.69 (t, C-21^§^), 30.02 (q, C-29), 28.14 (q, C-28), 27.95 (q, C-23), 27.13 (t, C-2), 26.25 (t, C-16), 26.25 (t, C-15), 23.38 (q, C-27), 18.49 (q, C-26), 17.32 (t, C-6), 16.23(q, C-25), 15.45 (q, C-24), 11.57 (q, C-1″).

### 4.9. N′-(3β-Acetoxy-11-oxo-18βH-olean-12-en-29-oyl)propionimidamide (**3b**)

Product **3b** (0.69 g, quantitative yield) was obtained as a white amorphous solid according to the general procedure **A** from compound **1** (0.61 g, 1.17 mmol), dry CH_2_Cl_2_ (20 mL), CDI (0.23 g, 1.40 mmol), and *N’*-hydroxypropionimidamide (0.12 g, 1.40 mmol). Crude product **3b** was purified by flash column chromatography (SiO_2_, CHCl_3_). Mp 120.0 °C [decomposition]. [$\alpha_{D}^{25}$] +95 (*c* 0.20 g/100 mL; CHCl_3_). HRMS: calc. for (C_35_H_54_O_5_N_2_)^+^ m/z = 582.4027; found m/z = 582.4026. ^1^H NMR (CDCl_3_, 400 MHz): *δ* = 5.59 (s, 1H, H-12), 4.98 (br.s, 2H, NH_2_), 4.47 (dd, 1H, *J_3a,2a_* = 11.6, *J_3a,2e_* = 4.8, H-3a), 2.73 (dm, 1H, *J_1e,1a_* = 13.6, *J = J =* 3.5 H-1e), 2.31 (ddd, 2H, *J = J = J =* 7.6, CH_2_-1′), 2.31 (s, 1H, H-9), 2.17 (dm, 1H, J=13.4, H-18), 2.08-1.94 (m, 5H; H-21, H-15a; 2.01 (s, 3H, CH_3_-32)), 1.90 (dm, 1H, H-19), 1.79 (m, 1H, H-16a), 1.74-1.50 (m, 5H; H-19′, H-2, H-7, H-6, H-2′), 1.50-1.27 (m, 8H; H-22, H-22′, H-7′, H-21′, H-6′, 1.33 (s, 3H, CH_3_-27)), 1.22 (s, 3H, CH_3_-29), 1.19 (dd, 3H, *J = J =* 7.7, CH3-2′′), 1.15–1.06 (m, 7H; H-16e; 1.11 (s, 3H, CH_3_-25), 1.08 (s, 3H, CH_3_-26), 1.05-0.94 (m, 2H; H-1a, H-15e), 0.84 (s, 6H, CH_3_-23, CH_3_-24), 0.77 (s, 3H, CH_3_-28), 0.75 (m, 1H, H-5a). ^13^C NMR (CDCl_3_, 100 MHz): *δ* = 200.07 (s, C-11), 172.71 (s, C-30), 170.90 (s, C-31), 169.47 (s, C-13), 159.70 (s, C-3′), 128.15 (d, C-12), 80.42 (d, C-3), 61.57 (d, C-9), 54.81 (d, C-5), 48.19 (d, C-18), 45.22 (s, C-14), 43.99 (s, C-20), 43.03 (s, C-8), 41.36 (t, C-19), 38.60 (t, C-1), 37.85 (s, C-4), 37.28 (t, C-22), 36.75 (s, C-10), 32.49 (t, C-7), 31.77 (s, C-17), 31.22 (t, C-21), 28.56 (q, C-28*), 28.08 (q, C-29*), 27.86 (q, C-23), 26.29 (t, C-16^§^), 26.19 (t, C-15^§^), 24.43 (t, C-1′′), 23.37 (t, C-2), 23.13 (q, C-27), 21.17 (q, C-32), 18.50 (q, C-26), 17.17 (t, C-6), 16.50 (q, C-24), 16.23 (q, C-25), 11.13 (q, C-2″).

### 4.10. 30-nor-3β-acetoxy-11-oxo-20-(3′-ethyl-1′,2′,4′-oxadiazol-5′-yl)-18βH-olean-12-en (**4b**)

Crude product **4b** (0.45 g, 67%) was obtained according to the general procedure **B** from compound **3b** (0.69 g, 1.17 mmol), Bu_4_NF 1M solution in THF (1.2 mL, 1.17 mmol), and THF (8 mL). Crude product **4b** was purified by flash column chromatography (SiO_2_, CHCl_3_) and recrystallized from MeOH–CH_2_Cl_2_ to give a pure sample of **4b** (0.38 g, 57%). Mp 279.5–280.6 °C. [$\alpha_{D}^{23}$] +163 (*c* 0.20 g/100 mL; CHCl_3_). HRMS: Calc. for (C_35_H_52_O_4_N_2_)^+^ m/z = 564.3922; found m/z = 564.3919. ^1^H NMR (CDCl_3_, 400 MHz): *δ* = 5.66 (s, 1H, H-12), 4.47 (dd, 1H, *J_3a,2a_* = 11.4, *J_3a,2e_* = 4.4, H-3a), 2.75 (dm, 1H, *J_1e,1a_* = 13.6, H-1e), 2.71 (ddd, 2H, *J = J = J =* 7.6, CH_2_-1″), 2.32 (s, 1H, H-9), 2.20-1.96 (m, 7H; H-21, H-18, H-19, H-15, 2.01 (s, 3H, CH_3_-32)), 1.84 (ddd, 1H, J = J = 14.03, H-19′), 1.78 (m, 1H, H-16), 1.73-1.49 (m, 5H; H-2, H-7, H-2′, H-21, H-6), 1.49-1.32 (m, 6H; H-6′, H-22, H-7′, 1.35 (s, 3H, CH_3_-27)), 1.32-0.95 (m, 16H; 1.28 (dd, 3H, *J = J =* 7.5, CH_3_-2′′), 1.26 (s, 3H, CH_3_-29), H-22′, H-16′, 1.11 (s, 3H, CH_3_-25), 1.07 (s, 3H, CH_3_-26), H-1a, H-15e), 0.83 (s, 6H, CH_3_-23, CH_3_-24), 0.76 (dm, 1H, H-5a), 0.71 (s, 3H, CH_3_-28). ^13^C NMR (CDCl_3_, 100 MHz): *δ* = 199.79 (s, C-11), 183.28 (s, C-5′(30)), 171.37 (s, C-3′), 170.83 (s, C-31), 168.39 (s, C-13), 128.56 (d, C-12), 80.41 (d, C-3), 61.55 (d, C-9), 54.82 (d, C-5), 47.30 (d, C-18), 45.23 (s, C-14), 42.98 (s, C-8), 41.41 (t, C-19), 38.58 (t, C-1), 38.31 (s, C-20), 37.86 (s, C-4), 36.93 (s, C-10), 36.73 (t, C-22), 32.50 (t, C-7), 31.75 (t, C-21), 31.64 (s, C-17), 29.91 (q, C-29), 28.15 (q, C-28), 27.86 (q, C-23), 26.21 (t, C-16, C-15), 23.37 (t, C-2), 23.26 (q, C-27), 21.16 (q, C-32), 19.68 (t, C-1″), 18.46 (q, C-26), 17.18 (t, C-6), 16.51 (q, C-24), 16.23 (q, C-25), 11.15 (q, C-2″).

### 4.11. 30-nor-3β-hydroxy-11-oxo-20-(3′-ethyl-1′,2′,4′-oxadiazol-5′-yl)-18βH-olean-12-en (**5b**)

Crude product **5b** (0.27 g, 85%) was obtained according to the general procedure **C** from compound **4b** (0.34 g, 0.60 mmol), KOH (0.20 g, 3.62 mmol), and MeOH (8 mL). Crude product **5b** was purified by flash column chromatography (SiO_2_, 0–1% MeOH in CHCl_3_) and recrystallized from AcOEt–CHCl_3_ to give pure sample of **5b** (0.15 g, 48%). Mp 273.1–274.2 °C (AcOEt). [$\alpha_{D}^{24}$] +191 (*c* 0.20 g/100 mL; CHCl_3_). HRMS: Calc. for (C_33_H_50_O_3_N_2_)^+^ m/z = 522.3816; found m/z = 522.3820. ^1^H NMR (CDCl_3_, 500 MHz): *δ* = 5.67 (s, 1H, H-12), 3.20 (m, 1H, H-3a), 2.77 (ddd, 1H, *J_1e,1a_* = 13.5, *J = J =* 3.5, H-1e), 3.04 (ddd, 2H, *J = J = J =* 7.6, CH_2_-1′′), 2.32 (s, 1H, H-9), 2.21-2.00 (m, 4H; H-21, H-18, H-19, H-15), 1.90-1.77 (m, 2H; H-19′, H-16), 1.69-1.53 (m, 5H; H-7, H-2, H-2′, H-6, H-21′), 1.45-1.36 (m, 6H; H-22, H-6′, H-7′, 1.37 (s, 3H, CH_3_-27)), 1.33-1.15 (m, 8H; 1.30 (dd, 3H, *J*=7.6, CH_3_-2′′), 1.27 (s, 3H, CH_3_-29), H-22′, H-16′), 1.11 (s, 3H, CH_3_-25), 1.09 (s, 3H, CH_3_-26), 1.05-0.91 (m, 5H; H-15′, 0.99 (s, 3H, CH_3_-23), 0.96 H-1′), 0.78 (s, 3H, CH_3_-24), 0.73 (s, 3H, CH_3_-28), 0.68 (dm, 1H, H-5). ^13^C NMR (CDCl_3_, 125 MHz): *δ* = 199.86 (s, C-11), 183.33 (s, C-5′(30)), 171.44 (s, C-3′), 168.24 (s, C-13), 128.72 (d, C-12), 78.66 (d, C-3), 61.74 (d, C-9), 54.88 (d, C-5), 47.38 (d, C-18), 45.30 (s, C-14), 43.09 (s, C-8), 41.61 (t, C-19), 39.03 (t, C-1), 39.03 (s, C-4), 38.37 (s, C-20), 37.03 (t, C-22), 37.03 (s, C-10), 32.69 (t, C-7), 31.86 (s, C-17*), 31.72 (t, C-21*), 29.94 (q, C-29), 28.21 (q, C-28), 28.00 (q, C-23), 27.23 (t, C-2), 26.35 (t, C-16, C-15), 23.38 (q, C-27), 19.72 (t, C-1′′), 18.57 (q, C-26), 17.40 (t, C-6), 16.23 (q, C-25), 15.46 (q, C-24), 11.20 (q, C-2″). 

### 4.12. N′-(3β-Acetoxy-11-oxo-18βH-olean-12-en-29-oyl)isobutyrimidamide (**3c**)

Product **3c** (2.55 g, quantitative yield) was obtained as a white amorphous solid according to the general procedure **A** from compound **1** (2.0 g, 3.9 mmol), dry CH_2_Cl_2_ (20 mL), CDI (0.76 g, 4.7 mmol), and *N’*-hydroxyisobutyrimidamide (0.48 g, 4.7 mmol). Crude product **3c** was purified by flash column chromatography (SiO_2_, CHCl_3_). Mp 120.3 °C [decomposition]. [$\alpha_{D}^{25}$] +87 (*c* 0.20 g/100 mL; CHCl_3_). HRMS: Calc. for (C_36_H_56_O_5_N_2_)^+^ m/z = 596.4184; found m/z = 596.4178. ^1^H NMR (CDCl_3_, 300 MHz): *δ* = 5.58 (s, 1H, H-12), 4.65 (br.s, 2H, NH_2_), 4.47 (dd, 1H, *J_3a,2a_* = 11.4, *J_3a,2e_* = 4.9, H-3a), 2.74 (ddd, 1H, *J_1e,1a_* = 13.6, *J = J =* 3.4, H-1e), 2.62 (m, 1H, *J* = 7.0, CH_2_-1′), 2.31 (s, 1H, H-9), 2.18 (dm, 1H, J=13.4, H-18), 2.09-1.94 (m, 5H; H-21, H-15a; 2.01 (s, 3H, CH_3_-32)), 1.90 (dm, 1H, H-19), 1.79 (m, 1H, H-16a), 1.74-1.50 (m, 5H; H-19′, H-2, H-7, H-6, H-2′), 1.50-1.29 (m, 8H; H-22, H-22′, H-7′, H-21′, H-6′, 1.33 (s, 3H, CH_3_-27)), 1.25-0.92 (m, 18H; 1.21 (s, 3H, CH_3_-29); 1.19 (d, 3H, *J* = 7.0, CH_3_-2′′); 1.18 (d, 3H, *J* = 7.0, CH_3_-3″); H-16e; 1.11 (s, 3H, CH_3_-25); 1.08 (s, 3H, CH_3_-26); H-1a, H-15e), 0.84 (s, 6H, CH_3_-23, CH_3_-24), 0.77 (s, 3H, CH_3_-28), 0.75 (m, 1H, H-5a). ^13^C NMR (CDCl_3_, 75 MHz): *δ* = 200.02 (s, C-11), 172.74 (s, C-30), 170.89 (s, C-31), 169.44 (s, C-13), 162.44 (s, C-3′), 128.18 (d, C-12), 80.43 (d, C-3), 61.58 (d, C-9), 54.82 (d, C-5), 48.20 (d, C-18), 45.22 (s, C-14), 44.02 (s, C-20), 43.04 (s, C-8), 41.37 (t, C-19), 38.61 (t, C-1), 37.85 (s, C-4), 37.29 (t, C-22), 36.76 (s, C-10), 32.51 (t, C-7), 31.78 (s, C-17), 31.24 (t, C-21), 30.70 (d, C-1′′), 28.66 (q, C-28*), 28.06 (q, C-29*), 27.87 (q, C-23), 26.30 (t, C-16^§^), 26.21 (t, C-15^§^), 23.38 (t, C-2), 23.15 (q, C-27), 21.17 (q, C-32), 20.06 (q, C-2″ ^#^), 20.01 (q, C-3′′ ^#^), 18.50 (q, C-26), 17.19 (t, C-6), 16.50 (q, C-24), 16.23 (q, C-25). 

### 4.13. 30-nor-3β-acetoxy-11-oxo-20-(3′-isopropyl -1′,2′,4′-oxadiazol-5′-yl)-18βH-olean-12-en (**4c**)

Crude product **4c** (2.26 g, quantitative yield) was obtained according to the general procedure **B** from compound **3c** (2.17 g, 3.64 mmol), Bu_4_NF 1M solution in THF (6.0 mL, 6.00 mmol), and THF (25 mL). Crude product **4c** was purified by flash column chromatography (SiO_2_, 0–4% MeOH gradient in CHCl_3_) and recrystallized from AcOEt to give a pure sample of **4c** (1.43 g, 68%). Mp 271.9–273.1 °C (AcOEt). [$\alpha_{D}^{21}$] +188 (*c* 0.20 g/100 mL; CHCl_3_). HRMS Calc. for (C_36_H_54_O_4_N_2_)^+^ m/z = 578.4078; found m/z = 578.4073. ^1^H NMR (CDCl_3_, 400 MHz): *δ* = 5.67 (s, 1H, H-12), 4.50 (dd, 1H, *J_3a,2a_* = 11.5, *J_3a,2e_* = 4.8, H-3a), 3.04 (m, 1H, *J* = 6.9, H-1′′), 2.78 (dm, 1H, *J_1e,1a_* = 13.6, H-1e), 2.35 (s, 1H, H-9), 2.23-1.94 (m, 7H; H-21, H-18, H-19, H-15, 2.03 (s, 3H, CH_3_-32)), 1.92-1.50 (m, 7H; H-19′, H-16, H-2, H-7, H-2′, H-21, H-6), 1.50-0.95 (m, 25H; H-6′, H-22, H-7′, 1.37 (s, 3H, CH_3_-27), 1.32 (s, 3H, CH_3_-2″ *), 1.30 (s, 3H, CH_3_-3″ *), 1.28 (s, 3H, CH_3_-29), H-22′, H-16′, 1.14 (s, 3H, CH_3_-25), 1.09 (s, 3H, CH_3_-26), H-1a, H-15e), 0.86 (s, 6H, CH_3_-23, CH_3_-24), 0.78 (dm, 1H, H-5a), 0.72 (s, 3H, CH_3_-28). ^13^C NMR (CDCl_3_, 125 MHz): *δ* = 199.84 (s, C-11), 183.13 (s, C-5′(30)), 174.90 (s, C-3′), 170.87 (s, C-31), 168.47 (s, C-13), 128.66 (d, C-12), 80.52 (d, C-3), 61.65 (d, C-9), 54.95 (d, C-5), 47.37 (d, C-18), 45.31 (s, C-14), 43.11 (s, C-8), 41.63 (t, C-19), 38.69 (t, C-1), 38.35 (s, C-20), 37.95 (s, C-4), 37.04 (s, C-10), 36.86 (t, C-22), 32.63 (t, C-7), 31.90 (t, C-21), 31.70 (s, C-17), 29.83 (q, C-29), 28.26 (q, C-28), 27.95 (q, C-23), 26.72 (d, C-1″), 26.35 (t, C-16^#^), 26.32 (t, C-15^#^), 23.47 (t, C-2), 23.30 (q, C-27), 21.17 (q, C-32), 20.36 (q, C-2″ ^§^), 20.36 (q, C-3″ ^§^), 18.57 (q, C-26), 17.29 (t, C-6), 16.57 (q, C-24), 16.28 (q, C-25).

### 4.14. 30-nor-3β-hydroxy-11-oxo-20-(3′-isopropyl-1′,2′,4′-oxadiazol-5′-yl)-18βH-olean-12-en (**5c**)

Crude product **5c** (0.86 g, quantitative yield) was obtained according to the general procedure **C** from compound **4c** (0.91 g, 1.57 mmol), KOH (0.53 g, 9.45 mmol), and MeOH (28 mL). Crude product **5c** was purified by flash column chromatography (SiO_2_, 0–2% MeOH in CHCl_3_) to give a white solid (0.67 g, 80%), then recrystallized from AcOEt to give a pure sample of **5c** (0.33 g, 39%). Mp 216.4–219.7 °C (AcOEt). [$\alpha_{D}^{23.5}$] +190 (*c* 0.20 g/100 mL; CHCl_3_). HRMS: Calc. for (C_34_H_52_O_3_N_2_)^+^ m/z = 536.3973; found m/z = 536.3975. ^1^H NMR (CDCl_3_, 500 MHz): *δ* = 5.67 (s, 1H, H-12), 3.20 (dd, 1H, *J_3a,2a_* = 10.9, *J_3a,2e_* = 5.3, H-3a), 3.04 (m, 1H, *J* = 6.9, H-1′′), 2.76 (ddd, 1H, *J_1e,1a_* = 13.5, *J = J =* 3.5, H-1e), 2.33 (s, 1H, H-9), 2.21-2.09 (m, 3H; H-21, H-18, H-19), 2.05 (m, 1H, H-15), 1.83 (dd, 1H, *J = J =* 13.6, H-19′), 1.82 (m, 1H, H-16), 1.70-1.54 (m, 5H; H-7, H-2, H-2′, H-6, H-21′), 1.48-1.36 (m, 6H; H-22, H-6′, H-7′, 1.38 (s, 3H, CH_3_-27)), 1.32 (s, 3H, CH_3_-2″ ^‡^), 1.31 (s, 3H, CH_3_-3″ ^‡^), 1.28 (s, 3H, CH_3_-29), 1.27-1.14 (m, 2H; H-22′, H-16′), 1.12 (s, 3H, CH_3_-25), 1.10 (s, 3H, CH_3_-26), 1.02 (m, 1H; H-15′), 0.99 (s, 3H, CH_3_-23), 0.96 (m, 1H, H-1′), 0.79 (s, 3H, CH_3_-24), 0.73 (s, 3H, CH_3_-28), 0.68 (dm, 1H, H-5). ^13^C NMR (CDCl_3_, 125 MHz): *δ* = 199.95 (s, C-11), 183.13 (s, C-5′(30)), 174.90 (s, C-3′), 168.39 (s, C-13), 128.73 (d, C-12), 78.69 (d, C-3), 61.76 (d, C-9), 54.89 (d, C-5), 47.36 (d, C-18), 45.31 (s, C-14), 43.11 (s, C-8), 41.68 (t, C-19), 39.04 (t, C-1), 39.04 (s, C-4), 38.35 (s, C-20), 37.04 (t, C-22*), 37.03 (s, C-10*), 32.70 (t, C-7), 31.92 (s, C-17^#^), 31.71 (t, C-21^#^), 29.84 (q, C-29), 28.28 (q, C-28), 28.01 (q, C-23), 27.25 (t, C-2), 26.73 (d, C-1′′), 26.37 (t, C-16^§^), 26.36 (t, C-15^§^), 23.37 (q, C-27), 30.38 (q, C-2′′ ^†^), 20.27 (q, C-3′′ ^†^), 18.59 (q, C-26), 17.41 (t, C-6), 16.24 (q, C-25), 15.47 (q, C-24).

### 4.15. N′-(3β-Acetoxy-11-oxo-18βH-olean-12-en-29-oyl)pivalimidamide (**3d**)

Product **3d** (1.8 g, quantitative yield) was obtained as a white amorphous solid according to the general procedure **A** from compound **1** (1.5 g, 2.9 mmol), dry CH_2_Cl_2_ (35 mL), CDI (0.73 g, 4.5 mmol), and *N’*-hydroxypivalimidamide (0.50 g, 4.3 mmol). Crude product **3d** was purified by flash column chromatography (SiO_2_, CHCl_3_). Mp 151.6–152.3 °C. [$\alpha_{D}^{21}$] +101 (*c* 0.20 g/100 mL; CHCl_3_). HRMS: Calc. for (C_37_H_58_O_5_N_2_)^+^ m/z = 610.4340; found m/z = 610.4346. ^1^H NMR (CDCl_3_, 500 MHz): *δ* = 5.59 (s, 1H, H-12), 4.79 (br.s, 2H, NH_2_), 4.46 (dd, 1H, *J_3a,2a_* = 11.7, *J_3a,2e_* = 4.6, H-3a), 2.74 (dm, 1H, *J_1e,1a_* = 13.6, H-1e), 2.30 (s, 1H, H-9), 2.20 (dm, 1H, J=13.4, H-18), 2.06-1.95 (m, 5H; H-21, H-15a; 2.00 (s, 3H, CH_3_-32)), 1.90 (dm, 1H, H-19), 1.79 (m, 1H, H-16a), 1.71-1.50 (m, 5H; H-19′, H-2, H-7, H-6, H-2′), 1.50-1.28 (m, 8H; H-22, H-22′, H-7′, H-21′, H-6′, 1.32 (s, 3H, CH_3_-27)), 1.23 (s, 9H, CH_3_-2′′, CH_3_-3′′, CH_3_-4′′), 1.21 (s, 3H, CH_3_-29), 1.14 (dm, 1H, H-16e), 1.11 (s, 3H, CH_3_-25), 1.08 (s, 3H, CH_3_-26), 1.04-0.94 (m, 2H; H-1a, H-15e), 0.83 (s, 6H, CH_3_-23, CH_3_-24), 0.78 (s, 3H, CH_3_-28), 0.75 (m, 1H, H-5a). ^13^C NMR (CDCl_3_, 125 MHz): *δ* = 199.87 (s, C-11), 172.65 (s, C-30), 170.77 (s, C-31), 169.27 (s, C-13), 164.67 (s, C-3′), 128.24 (d, C-12), 80.43 (d, C-3), 61.58 (d, C-9), 54.86 (d, C-5), 48.03 (d, C-18), 45.23 (s, C-14), 44.00 (s, C-20), 43.06 (s, C-8), 41.34 (t, C-19), 38.64 (t, C-1), 37.86 (s, C-4), 37.31 (t, C-22), 36.79 (s, C-10), 35.19 (d, C-1′′), 32.53 (t, C-7), 31.78 (s, C-17), 31.26 (t, C-21), 28.70 (q, C-28*), 28.01 (q, C-29*), 27.87 (q, C-23), 27.77 (q, C-2″, C-3″, C-4″), 26.31 (t, C-16^§^), 26.23 (t, C-15^§^), 23.39 (t, C-2), 23.18 (q, C-27), 21.10 (q, C-32), 18.53 (q, C-26), 17.21 (t, C-6), 16.48 (q, C-24), 16.20 (q, C-25).

### 4.16. 30-nor-3β-acetoxy-11-oxo-20-(3′-tert-butyl-1′,2′,4′-oxadiazol-5′-yl)-18βH-olean-12-en (**4d**)

Crude product **4d** (1.7 g, quantitative yield) was obtained according to the general procedure **B** from compound **3d** (1.7 g, 2.8 mmol), Bu_4_NF 1M solution in THF (2.6 mL, 2.6 mmol), and THF (20 mL). Crude product **4d** was purified by flash column chromatography (SiO_2_, CHCl_3_) and recrystallized from AcOEt to give a pure sample of **4d** (1.3 g, 76%). Mp 274.4–276.7 °C (EtOH). Mp 284.2–284.7 °C (AcOEt). [$\alpha_{D}^{21}$] +166 (*c* 0.20 g/100 mL; CHCl_3_). HRMS: Calc. for (C_37_H_56_O_4_N_2_)^+^ m/z = 592.4235; found m/z = 592.4243. ^1^H NMR (CDCl_3_, 400 MHz): *δ* = 5.66 (s, 1H, H-12), 4.50 (dd, 1H, *J_3a,2a_* = 11.5, *J_3a,2e_* = 4.5, H-3a), 2.78 (dm, 1H, *J_1e,1a_* = 12.9, H-1e), 2.35 (s, 1H, H-9), 2.25-1.96 (m, 7H; H-21, H-18, H-19, H-15, 2.03 (s, 3H, CH_3_-32)), 1.90-1.51 (m, 7H; H-19′, H-16, H-2, H-7, H-2′, H-21, H-6), 1.51-0.94 (m, 28H; H-6′, H-22, H-7′, 1.37 (s, 3H, CH_3_-27), 1.34 (s, 9H, CH_3_-2′′, CH_3_-3″, CH_3_-4″), 1.28 (s, 3H, CH_3_-29), H-22′, H-16′, 1.14 (s, 3H, CH_3_-25), 1.09 (s, 3H, CH_3_-26), H-1a, H-15e), 0.86 (s, 6H, CH_3_-23, CH_3_-24), 0.78 (dm, 1H, H-5a), 0.72 (s, 3H, CH_3_-28). ^13^C NMR (CDCl_3_, 125 MHz): *δ* = 199.91 (s, C-11), 182.86 (s, C-5′(30)), 177.46 (s, C-3′), 170.88 (s, C-31), 168.63 (s, C-13), 128.65 (d, C-12), 80.53 (d, C-3), 61.66 (d, C-9), 54.96 (d, C-5), 47.34 (d, C-18), 45.31 (s, C-14), 43.13 (s, C-8), 41.71 (t, C-19), 38.69 (t, C-1), 38.26 (s, C-20), 37.96 (s, C-4), 37.04 (s, C-10*), 36.88 (t, C-22*), 32.65 (t, C-7), 32.30 (d, C-1″), 31.95 (t, C-21^#^), 31.69 (s, C-17^#^), 29.70 (q, C-29), 28.32 (q, C-28), 28.26 (q, C-2′′, C-3″, C-4′′), 27.96 (q, C-23), 26.36 (t, C-16^§^), 26.33 (t, C-15^§^), 23.48 (t, C-2), 23.27 (q, C-27), 21.18 (q, C-32), 18.58 (q, C-26), 17.29 (t, C-6), 16.57 (q, C-24), 16.28 (q, C-25).

### 4.17. 30-nor-3β-hydroxy-11-oxo-20-(3′-tert-butyl -1′,2′,4′-oxadiazol-5′-yl)-18βH-olean-12-en (**5d**)

Crude product **5d** (0.26 g, 84%) was obtained according to the general procedure **C** from compound **4d** (0.33 g, 0.56 mmol), KOH (0.19 g, 3.34 mmol), and MeOH (8 mL). Crude product **5d** was purified by flash column chromatography (SiO_2_, CHCl_3_) to give a white amorphous solid (0.22 g, 73%). Mp 227.8 °C [decomposition]. [$\alpha_{D}^{24.5}$] +182 (*c* 0.20 g/100 mL; CHCl_3_). HRMS: Calc. for (C_35_H_54_O_3_N_2_)^+^ m/z = 550.4129; found m/z = 550.4132. ^1^H NMR (CDCl_3_, 400 MHz): *δ* = 5.66 (s, 1H, H-12), 3.21 (m, 1H, H-3a), 2.76 (dm, 1H, H-1e), 2.32 (s, 1H, H-9), 2.27-1.97 (m, 4H; H-21, H-18, H-19, H-15), 1.90-0.88 (m, 38H; H-19′, H-16, H-7, H-2, H-2′, H-6, H-21′, H-22, H-6′, H-7′, 1.37 (s, 3H, CH_3_-27), 1.33 (s, 9H, CH_3_-2″, CH_3_-3″, CH_3_-4″), 1.28 (s, 3H, CH_3_-29), H-22′, H-16′, 1.11 (s, 3H, CH_3_-25), 1.09 (s, 3H, CH_3_-26), H-15′, 0.98 (s, 3H, CH_3_-23), H-1′), 0.84-0.63 (m, 7H; 0.78 (s, 3H, CH_3_-24), 0.72 (s, 3H, CH_3_-28), H-5). ^13^C NMR (CDCl_3_, 125 MHz): *δ* = 199.99 (s, C-11), 182.85 (s, C-5′(30)), 171.45 (s, C-3′), 168.52 (s, C-13), 128.70 (d, C-12), 78.68 (d, C-3), 61.75 (d, C-9), 54.90 (d, C-5), 47.32 (d, C-18), 45.29 (s, C-14), 43.13 (s, C-8), 41.76 (t, C-19), 39.04 (t, C-1), 39.04 (s, C-4), 38.25 (s, C-20), 37.04 (t, C-22), 37.04 (s, C-10), 32.71 (t, C-7), 32.29 (s, C-1′′), 31.96 (s, C-17^#^), 31.8 (t, C-21^#^), 29.69 (q, C-29), 28.32 (q, C-28), 28.26 (q, C-2′′, C-3′′, C-4′′), 28.01 (q, C-23), 27.25 (t, C-2), 26.38 (t, C-16^§^), 26.35 (t, C-15^§^), 23.33 (q, C-27), 18.59 (q, C-26), 17.41 (t, C-6), 16.23 (q, C-25), 15.46 (q, C-24).

### 4.18. N′-(3β-Acetoxy-11-oxo-18βH-olean-12-en-29-oyl)benzimidamide (**3e**)

Crude product **3e** (1.9 g, quantitative yield) was obtained according to the general **A** from compound **1** (1.5 g, 2.9 mmol), dry CH_2_Cl_2_ (35 mL), CDI (0.73 g, 4.5 mmol), and *N’*-hydroxyphenylimidamide (0.61 g, 4.5 mmol). Crude product **3e** was purified by flash column chromatography (SiO_2_, CH_2_Cl_2_ then CHCl_3_) to give a pure sample of **3e** (1.8 g, 95%). Mp 144.6–145.0 °C. [$\alpha_{D}^{23}$] +132 (*c* 0.20 g/100 mL; CHCl_3_). HRMS: Calc. for (C_39_H_54_O_5_N_2_)^+^ m/z = 630.4027; found m/z = 630.4021. ^1^H NMR (CDCl_3_, 500 MHz): *δ* = 7.70 (m, 2H; H-2′′, H-6′′), 7.46-7.35 (m, 3H; H-3″, H-4″, H-5″), 5.64 (s, 1H, H-12), 5.03 (s, NH_2_), 4.49 (dd, 1H, *J* = 11.7, *J*=4.6, H-3), 2.75 (dm, 1H, H-1e), 2.33 (s, 1H, H-9), 2.25 (dm, 1H, H-18), 2.13–1.94 (m, 6H; H-21, H-15a, H-19; 2.02 (s, 3H, CH_3_-32)), 1.87-1.34 (m, 14H; H-16e, H-19′, H-2, H-2′, H-6, H-6′, H-7, H-7′, H-21′, H-22, H-22′; 1.36 (s, 3H, CH_3_-27)), 1.28 (s, 3H, CH_3_-29), 1.18 (dm, 1H, H-16a), 1.13 (s, CH_3_-25), 1.10 (s, CH_3_-26), 1.07-0.97 (m, 2H; H-1a, H-15e), 0.85 (s, 6H, CH_3_-23, CH_3_-24), 0.80 (s, 3H, CH_3_-28), 0.78 (d, 1H, H-5a). ^13^C NMR (CDCl_3_, 125 MHz): *δ* = 199.87 (s, C-11), 172.53 (s, C-30), 170.85 (s, C-31), 169.22 (s, C-13), 156.47 (s, C-3′), 130.95 (s, C-1″), 130.87 (d, C-4″), 128.54 (d, C-3″, C-5′′), 128.34 (d, C-12), 126.60 (d, C-2″, C-6′′), 80.49 (d, C-3), 61.66 (d, C-9), 54.93 (d, C-5), 48.28 (d, C-18), 45.28 (s, C-14), 44.14 (s, C-20), 43.13 (s, C-8), 41.54 (t, C-19), 38.71 (t, C-1), 37.92 (s, C-4), 37.41 (t, C-22), 36.87 (s, C-10), 32.61 (t, C-7), 31.86 (s, C-17), 31.41 (t, C-21), 28.67 (q, C-28*), 28.13 (q, C-29*), 27.92 (q, C-23), 26.40 (t, C-16^§^), 26.33 (t, C-15^§^), 23.45 (t, C-2), 23.22 (q, C-27), 21.14 (q, C-32), 18.60 (q, C-26), 17.27 (t, C-6), 16.53 (q, C-24), 16.25 (q, C-25).

### 4.19. 30-nor-3β-Acetoxy-11-oxo-20-(3′-Phenyl-1′,2′,4′-Oxadiazol-5′-yl)-18βH-olean-12-en (**4e**)

Crude product **4e** (1.4 g, 85%) was obtained according to the general procedure **B** from compound **3e** (1.7 g, 2.7 mmol), Bu_4_NF 1M solution in THF (0.2 mL, 0.2 mmol), and THF (20 mL). Crude product **4e** was purified by flash column chromatography (SiO_2_, CH_2_Cl_2_ then CHCl_3_) to give a pure sample of **4e** (1.2 g, 73%). Mp 266.3 °C [decomposition]. [$\alpha_{D}^{23}$] +213 (*c* 0.20 g/100 mL; CHCl_3_). HRMS: Calc. for (C_39_H_52_O_4_N_2_)^+^ m/z = 612.3922; found m/z = 612.3929. ^1^H NMR (CDCl_3_, 400 MHz): *δ* = 8.11-8.04 (m, 2H; H-2′′, H-6″), 7.51-7.42 (m, 3H; H-3″, H-4′′, H-5″), 5.73 (s, 1H, H-12), 4.50 (dd, 1H, *J* = 11.6, *J*=4.8, H-3), 2.79 (dm, 1H, *J* = 13.6, H-1e), 2.37 (s, 1H, H-9), 2.34-2.18 (m, 3H; H-21, H-18, H-19), 2.07 (m, 1H, H-15a), 2.03 (s, 3H, CH_3_-32), 1.90 (dd, 1H, J = J = 13.5, H-19′), 1.81 (m, 1H, H-16e), 1.76-1.52 (m, 5H; H-2, H-2′, H-6, H-7, H-21′), 1.52-1.31 (m, 9H; H-6′, H-7′, H-22; 1.40 (s, 3H, CH_3_-27); 1.34 (s, 3H, CH_3_-29)), 1.31-0.99 (m, 10H; H-16a, H-22′; 1.14 (s, 3H, CH_3_-25); 1.10 (s, 3H, CH_3_-26); H-1a, H-15e), 0.86 (s, 6H, CH_3_-23, CH_3_-24), 0.79 (d, 1H, *J* = 10.5, H-5a), 0.73 (s, 3H, CH_3_-28). ^13^C NMR (CDCl_3_, 100 MHz): *δ* = 199.93 (s, C-11), 183.65 (s, C-5′(30)), 170.89 (s, C-31), 168.56 (s, C-13), 168.05 (s, C-3′), 130.97 (d, C-4′′), 128.70 (d, C-3″, C-5″), 128.67 (d, C-12), 127.33 (d, C-2″, C-6″), 126.84 (s, C-1″), 80.46 (d, C-3), 61.63 (d, C-9), 54.88 (d, C-5), 47.27 (d, C-18), 45.29 (s, C-14), 43.06 (s, C-8), 41.48 (t, C-19), 38.63 (t, C-1), 38.51 (s, C-20), 37.91 (s, C-4), 37.05 (s, C-10), 36.81 (t, C-22), 32.55 (t, C-7), 31.84 (s, C-17*), 31.72 (t, C-21*), 29.91 (q, C-29), 28.19 (q, C-28), 27.91 (q, C-23), 26.27 (t, C-16, C-15), 23.44 (t, C-2), 23.35 (q, C-27), 21.21 (q, C-32), 18.52 (q, C-26), 17.23 (t, C-6), 16.56 (q, C-24), 16.28 (q, C-25).

### 4.20. 30-nor-3β-hydroxy-11-oxo-20-(3′-phenyl-1′,2′,4′-oxadiazol-5′-yl)-18βH-olean-12-en (**5e**)

Crude product **5e** (0.14 g, 76%) was obtained according to the general procedure **C** from compound **4e** (0.20 g, 0.33 mmol), KOH (0.11 g, 0.20 mmol), and MeOH (5 mL). Crude product **5e** was purified by flash column chromatography (SiO_2_, CHCl_3_) to give a white amorphous solid (0.13 g, 72%), then recrystallized from AcOEt to give a pure sample of **5e** (0.10 g, 58%). Mp 231.7–232.1 °C. [$\alpha_{D}^{23}$] +206 (*c* 0.20 g/100 mL; CHCl_3_). HRMS: Calc. for (C_37_H_50_O_3_N_2_)^+^ m/z = 570.3816; found m/z = 570.3814. ^1^H NMR (CDCl_3_, 400 MHz): *δ* = 8.13-8.04 (m, 2H; H-2′′, H-6′′), 7.53-7.42 (m, 3H; H-3″, H-4″, H-5″), 5.74 (s, 1H, H-12), 3.28-3.16 (m, 1H, H-3), 2.79 (dm, 1H, *J* = 13.5, H-1e), 2.40-2.16 (m, 4H; H-9, H-21, H-18, H-19), 2.07 (m, 1H, H-15a), 1.91 (dd, 1H, J = J = 13.6, H-19′), 1.82 (m, 1H, H-16e), 1.71-1.53 (m, 5H; H-2, H-2′, H-6, H-7, H-21′), 1.52-0.92 (m, 22H; H-6′, H-7′, H-22; 1.40 (s, 3H, CH_3_-27); 1.35 (s, 3H, CH_3_-29); H-22′, H-16a; 1.12 (s, 3H, CH_3_-25); 1.10 (s, 3H, CH_3_-26); H-15e, H-1a; 0.99 (s, 3H, CH_3_-23)), 0.79 (s, 3H, CH_3_-24), 0.74 (s, 3H, CH_3_-28), 0.70 (d, 1H, *J* = 11.6, H-5a). ^13^C NMR (CDCl_3_, 125 MHz): *δ* = 199.96 (s, C-11), 183.68 (s, C-5′(30)), 168.40 (s, C-13), 168.08 (s, C-3′), 130.96 (d, C-4′′), 128.77 (d, C-12), 128.71 (d, C-3′′, C-5′′), 127.37 (d, C-2″, C-6″), 126.91 (s, C-1″), 78.66 (d, C-3), 61.78 (d, C-9), 54.89 (d, C-5), 47.30 (d, C-18), 45.31 (s, C-14), 43.12 (s, C-8), 41.61 (t, C-19), 39.04 (t, C-1), 39.04 (s, C-4), 38.53 (s, C-20), 37.10 (s, C-10*), 37.04 (t, C-22*), 32.68 (t, C-7), 31.91 (s, C-17^#^), 31.75 (t, C-21^#^), 29.89 (q, C-29), 28.20 (q, C-28), 28.01 (q, C-23), 27.24 (t, C-2), 26.35 (t, C-16, C-15), 23.43 (q, C-27), 18.59 (q, C-26), 17.40 (t, C-6), 16.23 (q, C-25), 15.46 (q, C-24).

### 4.21. N′-(3β-Acetoxy-11-oxo-18βH-olean-12-en-29-oyl)Picolinimidamide (**3f**)

Crude product **3f** (2.52 g, quantitative yield) was obtained according to the general **A** from compound **1** (2.05 g, 4.0 mmol), dry CH_2_Cl_2_ (35 mL), CDI (0.97 g, 6.0 mmol), and *N’*- hydroxypicolinimidamide (1.10 g, 6.0 mmol). Crude product **3f** was purified by flash column chromatography (SiO_2_, CH_2_Cl_2_ then CHCl_3_) to give a pure sample of **3f** (2.48 g, 98%). Mp 165.8 °C [decomposition]. [$\alpha_{D}^{23.5}$] +139 (*c* 0.20 g/100 mL; CHCl_3_). HRMS: Calc. for (C_38_H_53_O_5_N_3_)^+^ m/z = 631.3980; found m/z = 631.3975. ^1^H NMR (CDCl_3_, 400 MHz): *δ* = 8.58-8.50 (m, 1H, H-6′′), 8.21-8.12 (m, 1H, H-3″), 7.76-7.67 (m, 1H, H-4″), 7.38-7.30 (m, 1H, H-5″), 5.65 (s, 1H, H-12), 5.27 (s, NH_2_), 4.49 (dd, 1H, *J* = 11.6, *J*=4.7, H-3), 2.76 (dm, 1H, *J*=13.6, H-1e), 2.33 (s, 1H, H-9), 2.25 (dm, 1H, H-18), 2.14–1.96 (m, 6H; H-21, H-15a, H-19; 2.02 (s, 3H, CH_3_-32)), 1.87-1.33 (m, 14H; H-16e, H-19′, H-2, H-2′, H-6, H-6′, H-7, H-7′, H-21′, H-22, H-22′; 1.36 (s, 3H, CH_3_-27)), 1.29 (s, 3H, CH_3_-29), 1.21-0.97 (m, 9H; H-16a, 1.12 (s, 3H, CH_3_-25), 1.09 (s, 3H, CH_3_-26), H-1a, H-15e), 0.85 (s, 6H, CH_3_-23, CH_3_-24), 0.82-0.74 (m, 4H; 0.79 (s, 3H, CH_3_-28), H-5a). ^13^C NMR (CDCl_3_, 100 MHz): *δ* = 199.87 (s, C-11), 172.71 (s, C-30), 170.90 (s, C-31), 169.02 (s, C-13), 153.78 (s, C-3′), 148.27 (d, C-6″), 147.35 (s, C-2″), 136.58 (d, C-4″), 128.36 (d, C-12), 125.39 (d, C-3″), 121.24 (d, C-5″), 80.45 (d, C-3), 61.59 (d, C-9), 54.85 (d, C-5), 48.11 (d, C-18), 45.22 (s, C-14), 44.16 (s, C-20), 43.04 (s, C-8), 41.39 (t, C-19), 38.64 (t, C-1), 37.89 (s, C-4), 37.42 (t, C-22*), 36.78 (s, C-10*), 32.54 (t, C-7), 31.82 (s, C-17^#^), 31.35 (t, C-21^#^), 28.68 (q, C-29), 28.10 (q, C-28), 27.89 (q, C-23), 26.33 (t, C-16^§^), 26.27 (t, C-15^§^), 23.42 (t, C-2), 23.19 (q, C-27), 21.19 (q, C-32), 18.53 (q, C-26), 17.22 (t, C-6), 16.53 (q, C-24), 16.25 (q, C-25).

### 4.22. 30-nor-3β-acetoxy-11-oxo-20-(3′-(pyridin-2′′-yl)-1′,2′,4′-oxadiazol-5′-yl)-18βH-olean-12-en (**4f**)

Crude product **4f** (2.0 g, 90%) was obtained according to the general procedure **B** from compound **3f** (2.28 g, 3.61 mmol), Bu_4_NF 1M solution in THF (0.4 mL, 0.4 mmol), and THF (25 mL). Crude product **4f** was purified by flash column chromatography (SiO_2_, 0–2% MeOH in CH_2_Cl_2_) to give a pure sample of **4e** (1.63 g, 74%). Mp 293.5–297.9 °C. [$\alpha_{D}^{23}$] +213 (*c* 0.20 g/100 mL; CHCl_3_). HRMS: Calc. for (C_38_H_51_O_4_N_3_)^+^ m/z = 613.3874; found m/z = 613.3871. ^1^H NMR (CDCl_3_, 400 MHz): *δ* = 8.78-8.71 (m, 1H, H-6′′), 8.12-8.05 (m, 1H, H-3″), 7.85-7.76 (m, 1H, H-4″), 7.71-7.33 (m, 1H, H-5″), 5.68 (s, 1H, H-12), 4.46 (dd, 1H, *J* = 11.5, *J*=4.7, H-3), 2.75 (dm, 1H, *J* = 13.7, H-1e), 2.37-2.12 (m, 4H; H-9, H-21, H-18, H-19), 2.10-1.85 (m, 5H; H-15a, 2.02 (s, 3H, CH_3_-32), H-19′), 1.83-1.47 (m, 6H; H-16e, H-2, H-2′, H-6, H-7, H-21′), 1.46-0.96 (m, 19H; H-6′, H-7′, H-22; 1.36 (s, 3H, CH_3_-27); 1.33 (s, 3H, CH_3_-29), H-16a, H-22′, 1.10 (s, 3H, CH_3_-25), 1.05 (s, 3H, CH_3_-26), H-1a, H-15e), 0.82 (s, 6H, CH_3_-23, CH_3_-24), 0.76 (d, 1H, *J* = 10.7, H-5a), 0.69 (s, 3H, CH_3_-28). ^13^C NMR (CDCl_3_, 100 MHz): *δ* = 199.77 (s, C-11), 184.53 (s, C-5′(30)), 170.77 (s, C-31), 168.30 (s, C-13), 167.91 (s, C-3′), 150.18 (d, C-6″), 146.40 (s, C-2″), 136.88 (d, C-4″), 128.56 (d, C-12), 125.24 (d, C-3′′), 123.17 (d, C-5″), 80.34 (d, C-3), 61.52 (d, C-9), 54.77 (d, C-5), 47.39 (d, C-18), 45.19 (s, C-14), 42.95 (s, C-8), 41.26 (t, C-19), 38.68 (t, C-1), 38.53 (s, C-20), 37.81 (s, C-4), 36.98 (t, C-22*), 36.69 (s, C-10*), 32.43 (t, C-7), 31.65 (s, C-17), 31.65 (t, C-21), 29.94 (q, C-29), 28.05 (q, C-28), 27.82 (q, C-23), 26.15 (t, C-16, C-15), 23.33 (t, C-2), 23.26 (q, C-27), 21.12 (q, C-32), 18.42 (q, C-26), 17.13 (t, C-6), 16.47 (q, C-24), 16.19 (q, C-25).

### 4.23. 30-nor-3β-hydroxy-11-oxo-20-(3′-(pyridin-2′′-yl)-1′,2′,4′-oxadiazol-5′-yl)-18βH-olean-12-en (**5f**)

Crude product **5f** (1.2 g, 87%) was obtained according to the general procedure **C** from compound **4f** (1.49 g, 2.43 mmol), KOH (0.7 g, 0.20 mmol), and *i*-PrOH (50 mL). Crude product **5f** was purified by flash column chromatography (SiO_2_, 0–1% MeOH in CHCl_3_) to give a white amorphous solid (1.11 g, 80%). Mp 244.2 °C [decomposition]. [$\alpha_{D}^{25}$] +193 (*c* 0.20 g/100 mL; CHCl_3_). HRMS: Calc. for (C_36_H_49_O_3_N_3_)^+^ m/z = 571.3768; found m/z = 571.3774. ^1^H NMR (CDCl_3_, 400 MHz): *δ* = 8.81–8.76 (m, 1H, H-6″), 8.15–8.10 (m, 1H, H-3″), 7.89–7.82 (m, 1H, H-4″), 7.45–7.39 (m, 1H, H-5″), 5.70 (s, 1H, H-12), 3.20 (m, 1H, H-3), 2.76 (dm, 1H, *J* = 13.5, H-1e), 2.38–2.29 (m, 2H; H-21, H-9), 2.27-2.13 (m, 2H; H-18, H-19), 2.05 (ddd, 1H, *J = J =* 13.6, *J* = 4.3, H-15a), 1.91 (dd, 1H, *J = J =* 13.0, H-19′), 1.79 (m, 1H, H-16e), 1.69–1.52 (m, 5H; H-2, H-2′, H-6, H-7, H-21′), 1.47–1.32 (m, 9H; H-6′, H-7′, H-22; 1.38 (s, 3H, CH_3_-27); 1.34 (s, 3H, CH_3_-29)), 1.27 (m, 1H, H-22′), 1.18 (m, 1H, H-16a), 1.12-0.90 (m, 11H; 1.09 (s, 3H, CH_3_-25), 1.07 (s, 3H, CH_3_-26), H-15e, H-1a, 0.97 (s, 3H, CH_3_-23)), 0.76 (s, 3H, CH_3_-24), 0.71 (s, 3H, CH_3_-28), 0.68 (d, 1H, *J* = 11.7, H-5a). ^13^C NMR (CDCl_3_, 100 MHz): *δ* = 200.03 (s, C-11), 184.67 (s, C-5′(30)), 168.36 (s, C-13), 167.71 (s, C-3′), 149.99 (d, C-6′′), 146.10 (s, C-2″), 137.31 (d, C-4″), 128.64 (d, C-12), 125.42 (d, C-3″), 123.33 (d, C-5″), 78.52 (d, C-3), 61.65 (d, C-9), 54.70 (d, C-5), 47.39 (d, C-18), 45.23 (s, C-14), 42.98 (s, C-8), 41.29 (t, C-19), 38.95 (t, C-1*), 38.90 (s, C-4*), 38.74 (s, C-20), 37.01 (s, C-10^#^), 36.87 (t, C-22^#^), 32.51 (t, C-7), 31.69 (s, C-17), 31.69 (t, C-21), 30.00 (q, C-29), 28.10 (q, C-28), 27.92 (q, C-23), 27.09 (t, C-2), 26.18 (t, C-16, C-15), 23.38 (q, C-27), 18.44 (q, C-26), 17.27 (t, C-6), 16.20 (q, C-25), 15.43 (q, C-24).

### 4.24. N′-(3β-Acetoxy-11-oxo-18βH-olean-12-en-29-oyl)nicotinimidamide (**3g**)

Crude product **3g** (2.6 g, quantitative yield) was obtained according to the general **A** from compound **1** (2.05 g, 4.0 mmol), dry CH_2_Cl_2_ (35 mL), CDI (0.97 g, 6.0 mmol), and *N’*- hydroxynicotinimidamide (1.10 g, 6.0 mmol). Crude product **3g** was purified by flash column chromatography (SiO_2_, CH_2_Cl_2_ then CHCl_3_) to give a pure sample of **3g** (2.37 g, 94%). Mp 149.7 °C [decomposition]. [$\alpha_{D}^{24}$] +125 (*c* 0.20 g/100 mL; CHCl_3_). HRMS: Calc. for (C_38_H_53_O_5_N_3_)^+^ m/z = 631.3980; found m/z = 613.3868; calcd for [M–H_2_O]^+^ m/z = 613.3868 (C_38_H_51_O_4_N_3_)^+^. ^1^H NMR (CDCl_3_, 500 MHz): *δ* = 8.93-8.89 (m, 1H, H-2″), 8.69-8.64 (m, 1H, H-6″), 8.07-8.03 (m, 1H, H-4″), 7.35-7.27 (m, 1H, H-5′′), 5.63 (s, 1H, H-12), 5.20 (s, NH_2_), 4.48 (dd, 1H, *J* = 11.7, *J*=4.6, H-3), 2.75 (dm, 1H, *J*=13.6, H-1e), 2.34 (s, 1H, H-9), 2.23 (dm, 1H, H-18), 2.13–1.94 (m, 6H; H-21, 2.02 (s, 3H, CH_3_-32), H-15a, H-19), 1.87-1.33 (m, 14H; H-16e, H-19′, H-2, H-2′, H-6, H-6′, H-7, H-7′, H-21′, H-22, H-22′; 1.36 (s, 3H, CH_3_-27)), 1.29 (s, 3H, CH_3_-29), 1.18 (dm, 1H, H-16a), 1.13 (s, 3H, CH_3_-25), 1.10 (s, 3H, CH_3_-26), 1.07-0.97 (m, 2H; H-1a, H-15e), 0.85 (s, 6H, CH_3_-23, CH_3_-24), 0.81 (s, 3H, CH_3_-28), 0.78 (dm, 1H, H-5a). ^13^C NMR (CDCl_3_, 125 MHz): *δ* = 199.92 (s, C-11), 172.44 (s, C-30), 170.86 (s, C-31), 169.20 (s, C-13), 154.34 (s, C-3′), 151.86 (d, C-6′′),147.52 (d, C-2″), 134.51 (d, C-4″), 128.34 (d, C-12), 127.21 (s, C-3″), 123.31 (d, C-5″), 80.48 (d, C-3), 61.68 (d, C-9), 54.92 (d, C-5), 48.35 (d, C-18), 45.30 (s, C-14), 44.19 (s, C-20), 43.15 (s, C-8), 41.53 (t, C-19), 38.71 (t, C-1), 37.93 (s, C-4), 37.41 (t, C-22), 36.88 (s, C-10), 32.61 (t, C-7), 31.87 (s, C-17*), 31.41 (t, C-21*), 28.64 (q, C-29), 28.16 (q, C-28), 27.93 (q, C-23), 26.39 (t, C-16^§^), 26.32 (t, C-15^§^), 23.45 (t, C-2), 23.22 (q, C-27), 21.15 (q, C-32), 18.60 (q, C-26), 17.26 (t, C-6), 16.54 (q, C-24), 16.26 (q, C-25).

### 4.25. 30-nor-3β-acetoxy-11-oxo-20-(3′-(pyridin-3′′-yl)-1′,2′,4′-oxadiazol-5′-yl)-18βH-olean-12-en (**4g**)

Crude product **4g** (1.75 g, 90%) was obtained according to the general procedure **B** from compound **3g** (2.0 g, 3.17 mmol), Bu_4_NF 1M solution in THF (0.3 mL, 0.3 mmol), and THF (25 mL). Crude product **4g** was purified by flash column chromatography (SiO_2_, 0–2% MeOH in CHCl_3_) to give a pure sample of **4g** (1.46 g, 75%). Mp 253.2 °C [decomposition]. [$\alpha_{D}^{23.5}$] +201 (*c* 0.20 g/100 mL; CHCl_3_). HRMS: Calc. for (C_38_H_51_O_4_N_3_)^+^ m/z = 613.3874; found m/z = 613.3866. ^1^H NMR (CDCl_3_, 500 MHz): *δ* = 9.32-9.27 (m, 1H, H-2′′), 8.74-8.69 (m, 1H, H-6″), 8.36-8.31 (m, 1H, H-4″), 7.43-7.37 (m, 1H, H-5″), 5.70 (s, 1H, H-12), 4.50 (dd, 1H, *J* = 11.7, *J*=4.7, H-3), 2.75 (dm, 1H, *J* = 13.5, H-1e), 2.36 (s, 1H, H-9), 2.30-2.19 (m, 3H; H-21, H-18, H-19), 2.08 (m, 1H, H-15a), 2.03 (s, 3H, CH_3_-32), 1.93 (dd, 1H, J = J = 14.7, H-19′), 1.82 (m, 1H, H-16e), 1.74-1.54 (m, 5H; H-2, H-2′, H-6, H-7, H-21′), 1.49-1.37 (m, 6H; H-6′, H-7′, H-22; 1.40 (s, 3H, CH_3_-27)), 1.36 (s, 3H, CH_3_-29), 1.28 (m, 1H, H-16a), 1.19 (m, 1H, H-22′), 1.14 (s, 3H, CH_3_-25), 1.10 (s, 3H, CH_3_-26) 1.08-1.00 (m, 2H; H-1a, H-15e), 0.86 (s, 6H, CH_3_-23, CH_3_-24), 0.79 (d, 1H, *J* = 10.8, H-5a), 0.75 (s, 3H, CH_3_-28). ^13^C NMR (CDCl_3_, 125 MHz): *δ* = 199.76 (s, C-11), 184.37 (s, C-5′(30)), 170.85 (s, C-31), 168.20 (s, C-13), 166.23 (s, C-3′), 151.84 (d, C-6′′), 148.56 (d, C-2′′), 134.70 (d, C-4″), 128.77 (d, C-12), 123.51 (d, C-5″), 123.22 (s, C-3″), 80.50 (d, C-3), 61.70 (d, C-9), 54.97 (d, C-5), 47.43 (d, C-18), 45.33 (s, C-14), 43.13 (s, C-8), 41.53 (t, C-19), 38.71 (t, C-1*), 38.68 (s, C-4*), 37.96 (s, C-20), 37.08 (t, C-22^#^), 36.89 (s, C-10^#^), 32.63 (t, C-7), 31.89 (t, C-21^§^), 31.78 (s, C-17^§^), 29.95 (q, C-29), 28.21 (q, C-28), 27.95 (q, C-23), 26.34 (t, C-16^‡^), 26.32 (t, C-15^‡^), 23.48 (t, C-2), 23.39 (q, C-27), 21.17 (q, C-32), 18.59 (q, C-26), 17.29 (t, C-6), 16.57 (q, C-24), 16.29 (q, C-25).

### 4.26. 30-nor-3β-hydroxy-11-oxo-20-(3′-(pyridin-3′′-yl)-1′,2′,4′-oxadiazol-5′-yl)-18βH-olean-12-en (**5g**)

Crude product **5g** (0.94 g, 84%) was obtained according to the general procedure **C** from compound **4g** (1.20 g, 1.96 mmol), KOH (0.66 g, 11.74 mmol), and MeOH (20 mL). Crude product **5g** was purified by flash column chromatography (SiO_2_, 0–2% MeOH in CHCl_3_) to give a white amorphous solid (0.87 g, 78%). Mp 231.6 °C [decomposition]. [$\alpha_{D}^{24}$] +191 (*c* 0.20 g/100 mL; CHCl_3_). HRMS: Calc. for (C_36_H_49_O_3_N_3_)^+^ m/z = 571.3768; found m/z = 571.3772. ^1^H NMR (CDCl_3_, 400 MHz): *δ* = 9.35-9.22 (m, 1H1 H-2″), 8.77-8.64 (m, 1H, H-6″), 8.40-8.25 (m, 1H1 H-4″), 7.47-7.34 (m, 1H, H-5″), 5.69 (s, 1H, H-12), 3.20 (m, 1H, H-3), 2.77 (dm, 1H, H-1e), 2.33 (s, 1H, H-9), 2.30-2.14 (m, 3H; H-21, H-18, H-19), 2.13-2.00 (m, 1H, H-15a), 1.99-1.74 (m, 2H; H-19′, H-16e), 1.73-0.87 (m, 27H; H-2, H-2′, H-6, H-7, H-21′, H-6′, H-7′, H-22, 1.39 (s, 3H, CH_3_-27), 1.34 (s, 3H, CH_3_-29), H-22′, H-16a, 1.10 (s, 3H, CH_3_-25), 1.09 (s, 3H, CH_3_-26), H-15e, 0.98 (s, 3H, CH_3_-23), H-1a), 0.78 (s, 3H, CH_3_-24), 0.74 (s, 3H, CH_3_-28), 0.68 (m, 1H, H-5a). ^13^C NMR (CDCl_3_, 100 MHz): *δ* = 199.98 (s, C-11), 184.32 (s, C-5′(30)), 168.25 (s, C-13), 166.16 (s, C-3′), 151.78 (d, C-6″), 148.45 (d, C-2′′), 137.71 (d, C-4″), 128.72 (d, C-12), 123.52 (d, C-5″), 123.14 (s, C-3″), 78.56 (d, C-3), 61.72 (d, C-9), 54.78 (d, C-5), 47.32 (d, C-18), 45.26 (s, C-14), 43.04 (s, C-8), 41.43 (t, C-19), 39.00 (t, C-1*), 38.96 (s, C-4*), 38.62 (s, C-20), 37.01 (s, C-10^#^), 36.94 (t, C-22^#^), 32.58 (t, C-7), 31.80 (s, C-17^§^), 31.72 (t, C-21^§^), 29.94 (q, C-29), 28.17 (q, C-28), 27.96 (q, C-23), 27.15 (t, C-2), 26.24 (t, C-16, C-15), 23.41 (q, C-27), 18.50 (q, C-26), 17.32 (t, C-6), 16.22 (q, C-25), 15.45 (q, C-24).

### 4.27. N′-(3β-Acetoxy-11-oxo-18βH-olean-12-en-29-oyl)isonicotinimidamide (**3h**)

Crude product **3h** (2.1 g, quantitative yield) was obtained according to the general procedure **A** from compound **1** (1.5 g, 2.9 mmol), dry CH_2_Cl_2_ (20 mL), CDI (0.79 g, 4.9 mmol), and *N’*-hydroxyisonicotinimidamide (0.75 g, 5.47 mmol). Crude product **3h** was purified by flash column chromatography (SiO_2_, 0–4% MeOH in CHCl_3_) to give a pure sample of **3h** (1.8 g, 97%). Mp 161.3 °C [decomposition]. [$\alpha_{D}^{23}$] +127 (*c* 0.20 g/100 mL; CHCl_3_). HRMS: Calc. for (C_38_H_53_O_5_N_3_)^+^ m/z = 631.3980; found m/z = 613.3870 for [M-H_2_O]^+^; calc. for (C_38_H_51_O_4_N_3_)^+^ m/z = 613.3874. ^1^H NMR (CDCl_3_, 500 MHz): *δ* = 8.65 (m, 2H; H-2′′, H-6″), 7.59 (m, 2H; H-3″, H-5″), 5.61 (s, 1H, H-12), 5.19 (s, NH_2_), 4.48 (dd, 1H, *J* = 11.7, *J*=4.6, H-3), 2.75 (dm, 1H, *J*=13.6, H-1e), 2.33 (s, 1H, H-9), 2.20 (dd, 1H, *J* = 13.4, *J* = 3.2, H-18), 2.12–1.93 (m, 6H; H-21, H-15a, H-19; 2.01 (s, 3H, CH_3_-32)), 1.86-1.33 (m, 14H; H-16e, H-19′, H-2, H-2′, H-6, H-6′, H-7, H-7′, H-21′, H-22, H-22′; 1.36 (s, 3H, CH_3_-27)), 1.28 (s, 3H, CH_3_-29), 1.18 (dm, 1H, H-16a), 1.12 (s, 3H, CH_3_-25), 1.10 (s, 3H, CH_3_-26), 1.06-0.96 (m, 2H; H-1a, H-15e), 0.85 (s, 6H, CH_3_-23, CH_3_-24), 0.80 (s, 3H, CH_3_-28), 0.77 (d, 1H, *J* = 11.1, H-5a). ^13^C NMR (CDCl_3_, 125 MHz): *δ* = 199.90 (s, C-11), 172.34 (s, C-30), 170.87 (s, C-31), 169.17 (s, C-13), 154.26 (s, C-3′), 150.31 (d, C-2′′, C-6′′), 138.57 (d, C-4″), 128.31 (d, C-12), 120.64 (d, C-3″, C-5″), 80.48 (d, C-3), 61.68 (d, C-9), 54.92 (d, C-5), 48.36 (d, C-18), 45.29 (s, C-14), 44.18 (s, C-20), 43.14 (s, C-8), 41.51 (t, C-19), 38.70 (t, C-1), 37.91 (s, C-4), 37.37 (t, C-22), 36.86 (s, C-10), 32.59 (t, C-7), 31.86 (s, C-17), 31.39 (t, C-21), 28.60 (q, C-28*), 28.14 (q, C-29*), 27.91 (q, C-23), 26.37 (t, C-16^§^), 26.30 (t, C-15^§^), 23.44 (t, C-2), 23.21 (q, C-27), 21.14 (q, C-32), 18.59 (q, C-26), 17.25 (t, C-6), 16.52 (q, C-24), 16.24 (q, C-25).

### 4.28. 30-nor-3β-acetoxy-11-oxo-20-(3′-(pyridin-4″-yl)-1′,2′,4′-oxadiazol-5′-yl)-18βH-olean-12-en (**4h**)

Crude product **4h** (1.0 g, 92%) was obtained according to the general procedure **B** from compound **3h** (1.15 g, 1.82 mmol), Bu_4_NF 1M solution in THF (0.2 mL, 0.2 mmol), and THF (20 mL). Crude product **4h** was purified by flash column chromatography (SiO_2_, 0–20% AcOEt in CH_2_Cl_2_) to give a pure sample of **4h** (0.8 g, 71%). Mp 265.3–267.3 °C. [$\alpha_{D}^{23.5}$] +196 (*c* 0.20 g/100 mL; CHCl_3_). HRMS: Calc. for (C_38_H_51_O_4_N_3_)^+^ m/z = 613.3874; found m/z = 613.3871. ^1^H NMR (CDCl_3_, 500 MHz): *δ* = 8.75 (m, 2H; H-2″, H-6″), 7.93 (m, 2H; H-3″, H-5″), 5.71 (s, 1H, H-12), 4.50 (dd, 1H, *J* = 11.7, *J*=4.6, H-3), 2.79 (dm, 1H, *J* = 13.6, H-1e), 2.36 (s, 1H, H-9), 2.29-2.18 (m, 3H; H-21, H-18, H-19), 2.07 (m, 1H, H-15a), 2.02 (s, 3H, CH_3_-32), 1.93 (dd, 1H, J = J = 14.8, H-19′), 1.82 (m, 1H, H-16e), 1.75-1.53 (m, 5H; H-2, H-2′, H-6, H-7, H-21′), 1.49-1.32 (m, 9H, H-6′, H-7′, H-22; 1.40 (s, 3H, CH_3_-27); 1.35 (s, 3H, CH_3_-29)), 1.30-1.16 (m, 2H; H-16a, H-22′), 1.14 (s, 3H, CH_3_-25), 1.10 (s, 3H, CH_3_-26), 1.08-1.00 (m, 2H; H-1a, H-15e), 0.86 (s, 6H, CH_3_-23, CH_3_-24), 0.79 (d, 1H, *J* = 11.0, H-5a), 0.74 (s, 3H, CH_3_-28). ^13^C NMR (CDCl_3_, 125 MHz): *δ* = 199.77 (s, C-11), 184.73 (s, C-5′(30)), 170.82 (s, C-31), 168.16 (s, C-13), 166.62 (s, C-3′), 150.58 (d, C-2″, C-6″), 134.29 (d, C-4″), 128.76 (d, C-12), 121.16 (d, C-3″, C-5′′), 80.47 (d, C-3), 61.69 (d, C-9), 54.95 (d, C-5), 47.39 (d, C-18), 45.33 (s, C-14), 43.12 (s, C-8), 41.49 (t, C-19), 38.71 (t, C-1), 38.71 (s, C-20), 37.94 (s, C-4), 37.07 (s, C-10), 36.88 (t, C-22), 32.61 (t, C-7), 31.85 (s, C-17*), 31.77 (t, C-21*), 29.90 (q, C-29), 28.19 (q, C-28), 27.94 (q, C-23), 26.31 (t, C-16, C-15), 23.47 (t, C-2), 23.37 (q, C-27), 21.15 (q, C-32), 18.57 (q, C-26), 17.27 (t, C-6), 16.55 (q, C-24), 16.27 (q, C-25).

### 4.29. 30-nor-3β-hydroxy-11-oxo-20-(3′-(pyridin-4′′-yl)-1′,2′,4′-oxadiazol-5′-yl)-18βH-olean-12-en (**5h**)

Crude product **5h** (0.45 g, 76%) was obtained according to the general procedure **C** from compound **4h** (0.50 g, 0.82 mmol), KOH (0.27 g, 4.9 mmol), and MeOH (20 mL). Crude product **5h** was purified by flash column chromatography (SiO_2_, 0–4% MeOH in CHCl_3_) to give a white amorphous solid (0.35 g, 76%). Mp 129.5–129.9 °C. [$\alpha_{D}^{23}$] +198 (*c* 0.20 g/100 mL; CHCl_3_). HRMS: Calc. for (C_36_H_49_O_3_N_3_)^+^ m/z = 571.3768; found m/z = 571.3762. ^1^H NMR (CDCl_3_, 400 MHz): *δ* = 8.76 (m, 2H; H-2′′, H-6′′), 7.94 (m, 2H; H-3′′, H-5′′), 5.71 (s, 1H, H-12), 3.21 (m, 1H, H-3), 2.78 (dm, 1H, *J* = 13.5, H-1e), 2.34 (s, 1H, H-9), 2.31-2.16 (m, 3H; H-21, H-18, H-19), 2.07 (ddd, 1H, *J = J =* 13.5, *J* = 4.3, H-15a), 1.92 (dd, 1H, *J = J =* 14.6, H-19′), 1.87-1.76 (m, 1H, H-16e), 1.73-1.54 (m, 5H; H-2, H-2′, H-6, H-7, H-21′), 1.50-1.31 (m, 9H; H-6′, H-7′, H-22; 1.39 (s, 3H, CH_3_-27); 1.34 (s, 3H, CH_3_-29)), 1.29-1.15 (m, 2H; H-16a, H-22′), 1.15-0.91 (m, 11H; 1.11 (s, 3H, CH_3_-25), 1.09 (s, 3H, CH_3_-26), H-15e, H-1a, 0.98 (s, 3H, CH_3_-23), 0.78 (s, 3H, CH_3_-24), 0.73 (s, 3H, CH_3_-28), 0.68 (d, 1H, *J* = 11.6, H-5a). ^13^C NMR (CDCl_3_, 100 MHz): *δ* = 200.04 (s, C-11), 184.69 (s, C-5′(30)), 168.27 (s, C-13), 166.56 (s, C-3′), 150.53 (d, C-2″, C-6″), 134.24 (d, C-4″), 128.74 (d, C-12), 121.17 (d, C-3″, C-5″), 78.56 (d, C-3), 61.73 (d, C-9), 54.78 (d, C-5), 47.29 (d, C-18), 45.28 (s, C-14), 43.04 (s, C-8), 41.40 (t, C-19), 39.00 (t, C-1^§^), 38.96 (s, C-4^§^), 38.67 (s, C-20), 37.01 (s, C-10*), 36.94 (t, C-22*), 32.58 (t, C-7), 31.78 (s, C-17^#^), 31.72 (t, C-21^#^), 29.93 (q, C-29), 28.17 (q, C-28), 27.96 (q, C-23), 27.16 (t, C-2), 26.23 (t, C-16, C-15), 23.42 (q, C-27), 18.50 (q, C-26), 17.32 (t, C-6), 16.23 (q, C-25), 15.46 (q, C-24). 

### 4.30. Cell Cultures and GA Derivatives

Human cervical carcinoma HeLa and KB-3-1 cells, human duodenal carcinoma HuTu-80 cells, and human lung adenocarcinoma A549 cells were obtained from the Russian Culture Collection (Institute of Cytology, RAS, St. Petersburg, Russia). Melanoma B16-F10 (hereafter, melanoma B16) cells were obtained from the Cell Culture Bank of the Blokhin National Medical Oncology Research Center (Moscow, Russia). Nonmalignant fibroblasts hFF3 and RAW264.7 macrophages were kindly provided by Dr. Olga A. Koval (Institute of Chemical Biology and Fundamental Medicine, SB RAS, Novosibirsk, Russia) and Prof. Dmitry Kuprash (Engelhardt Institute of Molecular Biology, RAS, Moscow, Russia), respectively. The cells were cultured in DMEM (Sigma-Aldrich Inc., St. Louis, MO, USA) supplemented with 10% (*v*/*v*) heat-inactivated fetal bovine serum (FBS) (BioloT, St. Petersburg, Russia) and antibiotic–antimycotic solution (100 U/mL penicillin, 100 µg/mL streptomycin, 0.25 µg/mL amphotericin). In the case of RAW264.7 macrophages, glucose concentration was additionally increased up to 4.5 mg/mL. The cells were incubated at 37 °C in a humidified 5% CO_2_-containing air atmosphere (hereafter referred to as standard conditions). GA derivatives were dissolved in DMSO (stock solutions: 10 mM) and stored at −20 °C prior to use.

### 4.31. Mice

Female C57Bl/6 mice (25–27 g) and nonlinear female ICR mice (30–35 g) were provided by the Vivarium of Institute of Chemical Biology and Fundamental Medicine, Siberian Branch of the Russian Academy of Sciences (SB RAS) (Novosibirsk, Russia). Mice were kept at 5 mice per cage in a natural light regimen with free access to food and water. Experiments were carried out in accordance with the European Communities Council Directive 86/609/CEE. The experimental protocols were approved by the Committee on the Ethics of Animal Experiments at the Institute of Cytology and Genetics SB RAS (Novosibirsk, Russia) (protocol No. 51 from 23.05.2019).

### 4.32. Analysis of Cytotoxicity of GA Derivatives by MTT Assay

Cells were seeded in tetra- or pentaplicates in 96-well plates at a density of 10^4^ (tumor cells and hFF4 fibroblasts) or 10^5^ (RAW264.7 cells) cells per well. Cells were incubated under standard conditions for 24 h, followed by their treatment with the investigated derivatives (0.5–50 µM) or corresponding concentrations of DMSO (control) for a further 48 h (tumor cells and fibroblasts) or 24 h (RAW264.7 cells). At the end of incubation period, 10 µL of MTT solution was added to the cells (final concentration: 0.5 mg/mL) and plates were incubated for an additional 2 h under standard conditions. Then, the MTT-containing medium was aspirated and formazan crystals formed in living cells were solubilized with 100 µL of DMSO. The absorbance of each well was read at test and reference wavelengths of 570 and 620 nm, respectively, on a Multiscan RC plate reader (Thermo LabSystems, Helsinki, Finland). The IC_50_ values (the concentration of compounds leading to a decrease of A_570_ to 50% of A_570_^control^) were calculated by extrapolation of the dose–response curves. Hierarchical clustering of revealed IC_50_ values of investigated compounds was carried out with Euclidean distance using a Morpheus platform (https://software.broadinstitute.org/morpheus/), a bioinformatic tool for analysis of data in a matrix format. A heatmap of SI values was constructed by Microsoft Excel 2010.

### 4.33. Apoptosis Assay by Flow Cytometry

HeLa cells were seeded in 24-well plates at density of 5 × 10^4^ cells per well and incubated under standard conditions for 24 h. Next, the cells were treated by compound **5f** at 7 or 10 µM for a subsequent 48 h, followed by dissociation of cell monolayers into single cells by TrypLE Express (Gibco, Taastrup, Denmark) - and washing with cold PBS. Thereafter, cells were resuspended in binding buffer at concentration of 10^6^ cells/mL. After double staining with Annexin V-FITC and propidium iodide (PI) from an Annexin V-FITC Apoptosis Detection Kit (Millipore, Bedford, MA, USA), cells were analyzed using a NovoCyte Flow Cytometer (ACEA Biosciences Inc., San Diego, CA, USA). For each sample, 10,000 events were acquired.

### 4.34. Analysis of Mitochondrial Membrane Potential

HeLa cells were cultured in 24-well plates for 24 h under standard conditions and treated by **5f** at 7 or 10 µM for a subsequent 48 h. After the treatment, the cells were collected and resuspended at 10^6^ cells/mL in PBS-contained JC-1 (5 µg/mL). After 30 min of incubation under standard conditions, the cells were centrifuged, washed with PBS, and analyzed using the NovoCyte Flow Cytometer (ACEA Biosciences Inc., San Diego, CA, USA). In total, 10,000 events were acquired for each sample.

### 4.35. Metacaspases Activity Assay

The activity of caspases was determined by using a CaspACE^TM^ FITC-VAD-FMK (Promega, Madison, WI, USA). Briefly, HeLa cells were cultured in 24-well plates for 24 h under standard conditions and treated by **5f** at 7 or 10 µM for 48 h. The cells were collected and incubated with fluorophore-conjugated pancaspase inhibitor FITC-VAD-FMK in darkness for 30 min at room temperature. After incubation, the cells were centrifuged, washed twice with PBS, and analyzed by flow cytometry. For each sample, 10,000 events were acquired.

### 4.36. Executioner Caspases-3/7 Activity Assay

The activity of caspase-3/7 in HeLa cells was evaluated using a Caspase-Glo^®^ 3/7 assay kit (Promega, Madison, WI, USA) according to manufacturer’s instructions. Briefly, the cells were cultured in white-walled 96-well plates (seeding density: 10^4^ cells/well) for 24 h under standard conditions and then were treated by **5f** at 7 or 10 µM for 48 h. After incubation, 100 µL of Caspase-Glo^®^ 3/7 Reagent was added to each wells, the plate was carefully shaken on a plate mixer, then incubated for 30 min in darkness at room temperature. The luminescence was measured using a luminometer (CLARIOstar plate reader (BMG Labtech, Ortenberg, Germany)).

### 4.37. Colony-Forming Assay

HeLa cells were seeded in 96-well plates in tetraplicates at a density of 100 cells/well and treated by **5f** (0.25, 0.5 µM) for 14 days under standard conditions. Then cell colonies were fixed with 4% paraformaldehyde, stained with crystal violet dye (0.1% *w*/*v*), and photographed using a Vilber Lourmat transilluminator (Marne La Vallee, France). The percentage of area covered by cell colonies was calculated using a ColonyArea ImageJ plugin [91].

### 4.38. Measurement of Cell Motility Using xCelligence Technology

HeLa cells were seeded in tetraplicate at 2 × 10^4^ cells per well in the upper chamber of CIM-plates, in the presence or absence of **5f** at 0.5 µM. To stimulate cell migration, the lower chamber of the CIM-plate contained 10% FBS as chemoattractant. The cell index (electrical impedance) was monitored by the Real-Time Cell Analyzer Dual Plates (RTCA DP) xCelligence System (ACEA Biosciences, San Diego, CA, USA) every 1 h for 48 h.

### 4.39. Tumor Transplantation and Design of Animal Experiments

To generate a metastatic model of tumor progression, 10^5^ B16 melanoma cells suspended in 0.2 mL of saline buffer were inoculated into the lateral tail vein of C57Bl/6 mice. On day 4 after tumor transplantation, mice were divided into three groups (n = 9 per group): (1) mice without treatment (control); (2) mice who received intraperitoneal (i.p.) injections of 10% Tween-80 (vehicle); and (3) mice who received i.p. injections of **5f** in 10% Tween-80 at a dose of 50 mg/kg. The treatment was carried out three times a week. The total number of injections was five. Mice were sacrificed on day 14 after B16 melanoma transplantation and the lungs were collected for calculation of surface metastases.

### 4.40. Toxicity Assessment

During the experiment, general status of the animals and body weight were monitored. At the end of the experiment, organs were collected and organ indexes were calculated as (organ weight / body weight) × 100%.

### 4.41. Analysis of the Number of Metastases and Metastasis Inhibition Index

To study the antimetastatic action of **5f**, evaluation of surface metastases in lungs of B16 melanoma bearing mice was carried out. Surface metastases in the lungs were counted using a binocular microscope. Inhibition of metastases development was assessed using the metastasis inhibition index (MII), calculated according to the formula:$$\mathrm{MII}=\frac{\left(\sum_{n}\frac{\left(\bar{N}control-Nn\right)\times 100\%\;}{\bar{N}control}\right)}{n},$$
where *N* is the number of surface metastases and *n* is the number of mice in the analyzed group. The MII of the control group was taken as 0% and the MII reflecting the absence of metastases was taken as 100%.

### 4.42. In Silico Prediction of the Anti-Inflammatory Potential of the Tested Compounds

The probable anti-inflammatory activity of novel GA derivatives was predicted by PASS Online tool (http://www.pharmaexpert.ru/passonline/). During the analysis, only inflammatory-related terms “anti-inflammatory” and “nitric oxide antagonist” were selected. Bioactivities predicted by PASS Online tool were characterized by the probabilities of tested compounds to be active P_a_ or inactive P_i_. Heatmap of revealed PASS Score (P_a_/P_i_) of GA derivatives was constructed using the Morpheus platform (https://software.broadinstitute.org/morpheus/).

### 4.43. The Measurement of TNF-α and NO Production by Inflamed Macrophages

RAW264.7 cells were seeded in tetraplicates in 96-well plates at a density of 10^5^ cells/well and incubated under standard conditions for 24 h. The cells were treated with murine IFNγ (20 ng/mL) (Gibco, Grand Island, NY, USA) with or without (control) **3d** (10–50 µM) for a further 24 h. At the end of the incubation period, the levels of TNF-α and NO in culture medium were measured using murine TNF alpha ELISA (Thermo Scientific, Frederick, MD, USA) and the Griess reagent system (Promega, Madison, WI, USA), respectively, according to manufacturers’ instructions.

### 4.44. Carrageenan-Induced Paw Edema

To induce paw edema, ICR mice were injected with 1% carrageenan (50 μL) on the plantar side of the right hind paw. Compound **3d** (50 mg/kg), vehicle (sesame oil), and indomethacin (20 mg/kg) as a reference drug with proved anti-inflammatory effect were administered via gastric gavage 1 h before phlogogen injection. Mice were sacrificed 5 h after induction of paw edema. The percent of edema weight in carrageenan-injected compound-treated groups in comparison to that in vehicle-treated control group was calculated according to the formula:$$\%\;\mathrm{Edema}=\frac{{\left(\frac{M_{1}-M_{2}}{M_{2}}\right)}^{compound}}{\sum_{n}{\left(\frac{M_{1}-M_{2}}{M_{2}}\right)}^{vehicle}}\times n\times 100\%,$$
where *M_1_* is the weight of the paw with carrageenan-induced edema; *M_2_* is the weight of the intact paws; *compound* and *vehicle* correspond to the compound- and vehicle-treated phlogogen-injected mice, respectively; and *n* is the number of mice in the control group (n = 9). Furthermore, paw material was collected for subsequent histological analysis.

### 4.45. Histology

For histological analysis, the specimens of intact and edema paws from each animal were dehydrated in ascending ethanol and xylol and embedded in HistoMix paraffin (BioVitrum, Russia). Paraffin sections (5 μm) were sliced on a Microm HM 355S microtome (Thermo Fisher Scientific, Waltham, MA, USA) and stained with hematoxylin and eosin. Images were obtained using an Axiostar Plus microscope equipped with an Axiocam MRc5 digital camera (Zeiss, Oberkochen, Germany) at ×100 and ×400 magnifications.

### 4.46. Carrageenan-Induced Peritonitis

To study the effect of **3d** on carrageenan-induced peritonitis, ICR mice were injected with 1% carrageenan (250 μL) intraperitoneally (i.p.). Compound **3d** (50 mg/kg), vehicle (10% Tween-80), dexamethasone (1 mg/kg) as a reference anti-inflammatory drug, and saline buffer were administered i.p. 1 h before peritonitis induction. The healthy uninflamed group (control) was injected i.p. by saline. Mice were sacrificed 4 h after carrageenan injection and their peritoneal cavities were washed with 2 mL of heparinized cold saline buffer to obtain peritoneal cells. The peritoneal exudates were collected and the total leukocyte counts were performed with a Neubauer chamber by optical microscopy after diluting the peritoneal fluid with Turk solution (1:20). To determine the differential leukocyte counts (total 100 cells), peritoneal cells were centrifuged and placed onto slides, stained with azur–eosin by Romanovsky–Giemsa and examined by optical microscopy. The results were expressed as the number of total leukocytes (×10^5^ cells/mL) and the percentages of subpopulations of neutrophils, lymphocytes, and monocytes.

### 4.47. Prediction of Probable Primary Targets of **3d**

The probable primary protein targets of **3d** were predicted by Polypharmacology Browser 2.0 (http://ppb2.gdb.tools/), using the ECfp4 Naïve–Bayes machine learning model produced on the fly with 2000 nearest neighbors from the extended connectivity fingerprint of ECfp4 (NN(ECfp4)+NB(ECfp4)), according to the developer’s instructions [75].

### 4.48. PPI Network Reconstruction

The protein–protein interactions between revealed probable primary targets of **3d** and key genes associated with rodent inflammatome, which were identified previously by Wang et al. on 11 independent rodent inflammatory disease models [76], were predicted based on data deposited in the Search Tool for Retrieval of Interacting Genes/Proteins (STRING) database [92] with a confidence score >0.7. The reconstructed protein–protein pairs included functional relationships of proteins from five sources: published high-throughput experiments, genomic context prediction, co-expression, automated text mining, and PPI deposited in other databases. The created PPI network was visualized by Cytoscape 3.6.1. To compute the level of interconnection of **3d**’s probable targets in the network, node degree scores were calculated using the NetworkAnalyzer plugin [93].

### 4.49. Molecular Docking

Docking of **3d** with MMP9 (Protein Data Bank (PDB) ID: 1GKC), neutrophil elastase (PDB ID: 5A09), and thrombin (PDB ID: 2ZDA) was performed using Autodock Vina [94]. The three-dimensional structures of the mentioned proteins were uploaded from The Research Collaboratory for Structural Bioinformatics (RCSB) Protein Data Bank (https://www.rcsb.org/), followed by extraction of co-crystalized ligands from the PDB files of proteins, then the addition of polar hydrogen and Gasteiger charges into the protein structure using AutoDock Tools 1.5.7. The 2D structure of **3d** was converted to 3D form and its geometry was optimized with the MMFF94 force field using Marvin Sketch 5.12 and Avogadro 1.2.0, respectively. All rotatable bonds within the ligand were allowed to rotate freely. The docking parameters were set as follows: MMP9, center_x = 65.543, center_y = 31.351, center_z = 117.764, size_x = 22, size_y = 20, size_z = 30; neutrophil elastase (ELANE), center_x = -6.852, center_y = 36.233, center_z = -4.224, size_x = 22, size_y = 22, size_z = 30; thrombin (F2), center_x = 15.88, center_y = -13.212, center_z = 22.865, size_x = 34, size_y = 20, size_z = 16. The best molecular interactions were identified based on the binding orientation of the proteins’ key residues and their corresponding binding energy values. The results were imported and analyzed using Discovery Studio Visualizer 17.2.0. The 2D plots of the protein–ligand interactions were analyzed using LigPlot+ 1.4.5.

## 5. Conclusions

The performed cytotoxicity screening of novel 3′-substituted-1′,2′,4′-oxadiazole-containing GA derivatives and their intermediates revealed the hit compound (**5f**), bearing 3ʹ-pyridin-2″-yl substituent, which displays high antitumor selectivity and complex effects on HeLa cervical carcinoma cells, including the inhibition of their clonogenicity and motility, and a triggering of the intrinsic apoptotic pathway. The antitumor potential of **5f** was further validated in vivo; the investigated compound effectively inhibited the metastatic development of highly aggressive B16 melanoma in mice. Additionally, comprehensive in silico analysis of synthesized GA derivatives revealed intermediate **3d**—containing the *tert*-butyl group in the *O*-acylated amidoxime moiety—as a promising anti-inflammatory candidate able to bind to active sites of MMP9, neutrophil elastase, and thrombin. The performed experiments on cellular and murine models clearly confirmed the obtained computational data; it was found that **3d** effectively inhibited inflammatory response in RAW264.7 macrophages in vitro and carrageenan-induced inflammation in mice. Altogether, our findings provide new insights into the structure–activity relationship of heterocycle-containing triterpenoids and demonstrate that both oxadiazole-bearing compounds and their intermediates can display promising bioactivities.

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
