# Peer review of "Novel 3′-Substituted-1′,2′,4′-Oxadiazole Derivatives of 18βH-Glycyrrhetinic Acid and Their O-Acylated Amidoximes: Synthesis and Evaluation of Antitumor and Anti-Inflammatory Potential In Vitro and In Vivo"

_ijms, 2020, doi:10.3390/ijms21103511_

Round 1
Reviewer 1 Report
Although this manuscript includes a huge amount of data, the description and justification are not systematically organized very well. Furthermore, the significant improvements were not observed in the current study. This reviewer basically recommend the manuscript for publication in the International Journal of Molecular Sciences, however, the following issues should be considered before acceptance:
1) The readability of the manuscript would likely benefit from rewriting the introduction part. The drug design rationale is poor in the current form. Authors should add a figure with structures of bioactive oxadiazoles and explain how they arrived to the design of compounds 4 and 5.
2) The manuscript regroups a great amount of research work. However, I am surprised to see that in vivo experiments were conducted on compounds with micromolar IC50 (not very active) and without performing ADMET experiments (clearance, half-time life, etc). Authors should add the approval number of the animal welfare committee or indicate an institutional review for the safety (according to the ethical guidelines)!
3) Line 243: change lead with optimum hit or optimized hit. A lead compound will undergo preclinical and clinical studies. Compound 5f with micromolar efficiency is not a lead.
4) English must be corrected. Errors start from the title "3ʹ-subsituted'' must be changed with ''3'-substituted". Authors should use pas tense to present the results.
5) In scheme 1, authors should add final yields for compounds. In the legend, for example CH2Cl2 (numbers should be subscripts), etc. Compound numbers should be written in bold, etc.
6) MMP9 is an anticancer biological target. I didn't understand the link with the antiinflammatory activity of compound 3d. It seems an indirect effect. Authors should better explain this.
Author Response
Dear Reviewer #1,
We are sincerely thankful to you for your deep analysis of our manuscript and highly valuable remarks. We revised the manuscript according to your comments and, please, let us respond to your questions.
1) The readability of the manuscript would likely benefit from rewriting the introduction part. The drug design rationale is poor in the current form. Authors should add a figure with structures of bioactive oxadiazoles and explain how they arrived to the design of compounds 4 and 5.
Authors: Corrected. The Introduction part was modified: (a) Figure 1, containing the structures of mentioned bioactive oxadiazole-bearing derivatives of natural compounds and their O-acylamidoxime intermediates with their IC50 values, was added; (b) the description of the rationale of introduction of oxadiazole moieties and their O-acylamidoxime precursor groups into the natural metabolites was reinforced by additional references describing their improving effect on bioactivity of the parent molecules (lines 60-64, marked by yellow) and their ability to reorganize the binding mode of the derivatives with biologically relevant protein targets (lines 56-57, marked by yellow). Moreover, some favorable properties of pentacyclic triterpenoids were additionally added to emphasize the excellent suitability of these compounds as a starting material (lines 66-68, marked by yellow).
2) The manuscript regroups a great amount of research work. However, I am surprised to see that in vivo experiments were conducted on compounds with micromolar IC50 (not very active) and without performing ADMET experiments (clearance, half-time life, etc). Authors should add the approval number of the animal welfare committee or indicate an institutional review for the safety (according to the ethical guidelines)!
The major task of our study was to assess antitumor and anti-inflammatory potential of novel oxadiazole-bearing GA derivatives and their intermediates in vitro and in silico with subsequent validation of bioactivity of hit compounds in murine models. Their full-scale preclinical evaluation and ADMET experiments were not the aim of current research.
In our work, the hit cytotoxic derivative 5f was identified based not directly on its IC50 values, but on its highest selectivity of cytotoxic effect in tumor cells in comparison with non-malignant fibroblasts (SI5f(HeLa) > 13.9). According to published data, the micromolar IC50 values of triterpenoid derivatives should not be considered as a limiting factor in evaluation of their bioactivities in murine models – previously, triterpenoid derivatives with micromolar IC50 values had been selected for antitumor evaluation in vivo in the reports of other research groups, including investigations of CDDO (IC50 = 3.7 µM) and CDDO-Me (IC50 = 1 – 5 µM) in PC3 and MCF-7 xenografts, respectively [1,2], 23-hydroxybetulinic acid derivative (IC50 = 2.9 µM) in melanoma B16 model [3], Pulsatilla saponin D derivative (IC50 = 1.7 – 4.5 uM) in H22 hepatocellular carcinoma [4], etc.
It should be noted, that all animal works were performed by us according to the European Communities Council Directive 86/609/CEE and was approved by the Committee on the Ethics of Animal Experiments at the Institute of Cytology and Genetics SB RAS (Novosibirsk, Russia) (protocol #51 from 23.05.2019) (please, see lines 1118-1120 in Material and Methods).
3) Line 243: change lead with optimum hit or optimized hit. A lead compound will undergo preclinical and clinical studies. Compound 5f with micromolar efficiency is not a lead.
Authors: Corrected. Thank you for the explanation of the term ‘lead’. The ‘lead’ and ‘lead compound’ were replaced with ‘hit’ and ‘hit compounds’ throughout the manuscript (please, see lines 27, 35, 83, 215, 220, 256, 477, 608; marked by yellow).
4) English must be corrected. Errors start from the title "3ʹ-subsituted'' must be changed with ''3'-substituted". Authors should use pas tense to present the results.
Authors: Corrected. The current manuscript had been undergone English editing by MDPI English Editing Services. Please, find attached certificate, confirming the editing.
5) In scheme 1, authors should add final yields for compounds. In the legend, for example CH2Cl2 (numbers should be subscripts), etc. Compound numbers should be written in bold, etc.
Authors: Corrected. It is our disappointing misprints. The final yields for synthesized derivatives were added to Scheme 1; the compound numbers and numbers in chemical formulas were rewritten in bold and subscript mode, respectively.
6) MMP9 is an anticancer biological target. I didn't understand the link with the antiinflammatory activity of compound 3d. It seems an indirect effect. Authors should better explain this.
Authors: Corrected. Indeed, matrix metalloproteinase MMP9 is more known as important regulator of metastasis, angiogenesis and a key antitumor target. However, MMP9 can also participate in regulation of inflammation basically due to its stimulating effects on neutrophil migration to inflamed sites. MMP9 can process (i.e. activate) interleukin-8, a key chemotactic factor for neutrophils [5], and, moreover, can control the creation of chemokine gradient, which directs the migration of inflammatory cells to inflammatory regions [6]. The usage of MMP9 inhibitors was found to suppress inflammation in vivo in a range of models [7,8]. In order to more clearly show the linkage between MMP9 and inflammation, some corrections were introduced into the paragraph describing this interconnection (please, see lines 585-586; marked by yellow).
References:
- Deeb, D.; Gao, X.; Jiang, H.; Dulchavsky, S.A.; Gautam, S.C. Oleanane triterpenoid CDDO-Me inhibits growth and induces apoptosis in prostate cancer cells by independently targeting pro-survival Akt and mTOR. Prostate 2009, 69, 851–860.
- Konopleva, M.; Zhang, W.; Shi, Y.X.; McQueen, T.; Tsao, T.; Abdelrahim, M.; Munsell, M.F.; Johansen, M.; Yu, D.; Madden, T.; et al. Synthetic triterpenoid 2-cyano-3,12-dioxooleana-1,9-dien-28-oic acid induces growth arrest in HER2-overexpressing breast cancer cells. Mol. Cancer Ther. 2006, 5, 317–328.
- Lu, L.; Zhang, H.; Liu, J.; Liu, Y.; Wang, Y.; Xu, S.; Zhu, Z.; Xu, J. Synthesis, biological evaluation and mechanism studies of C-23 modified 23-hydroxybetulinic acid derivatives as anticancer agents. Eur. J. Med. Chem. 2019, 182, 111659.
- Fang, Y.; Hu, D.; Li, H.; Hu, J.; Liu, Y.; Li, Z.; Xu, G.; Chen, L.; Jin, Y.; Yang, S.; et al. Synthesis, biological evaluation, and mode of action of Pulsatilla saponin D derivatives as promising anticancer agents. Front. Pharmacol. 2019, 10.
- Ma, L.; Dorling, A. The roles of thrombin and protease-activated receptors in inflammation. Semin. Immunopathol. 2012, 34, 63–72.
- Manon-Jensen, T.; Multhaupt, H.A.B.; Couchman, J.R. Mapping of matrix metalloproteinase cleavage sites on syndecan-1 and syndecan-4 ectodomains. FEBS J. 2013, 280, 2320–2331.
- Moore, B.A.; Manthey, C.L.; Johnson, D.L.; Bauer, A.J. Matrix metalloproteinase-9 inhibition reduces inflammation and improves motility in murine models of postoperative ileus. Gastroenterology 2011, 141, 1283–1292.
- Corry, D.B.; Kiss, A.; Song, L.-Z.; Song, L.; Xu, J.; Lee, S.-H.; Werb, Z.; Kheradmand, F. Overlapping and independent contributions of MMP2 and MMP9 to lung allergic inflammatory cell egression through decreased CC chemokines. FASEB J. 2004, 18, 995–997.

Reviewer 2 Report
The article entitled: Novel 3ʹ-subsituted-1ʹ,2ʹ,4ʹ-oxadiazole derivatives of 18βH-glycyrrhetinic acid and their O-acylated amidooximes: synthesis and evaluation of anti-tumor and anti-inflammatory potential in vitro and in vivo is very well written. The paper includes synthesis and biological evaluation of a series of novel 18βH-glycyrrhetinic acid (GA) derivatives containing oxadiazole moieties. The authors systematically analyzed all prepared compounds starting from cytotoxicity of prepared compounds, to find the best selective antitumor derivative for further investigation. In silico investigation revealed an intermediate compound as a potential anti-inflammatory candidate, and further experiments, as well as docking simulations, confirmed this assumption. The investigation is very well conducted, and every step is explained.
In my opinion, this paper is acceptable for the publication in “International Journal of Molecular Sciences” after minor revisions such as typos listed below:
In the legend of Scheme 1:
- Instead of CH2Cl2 it should be formula stile CH2Cl2
- Instead of R-C(=NOH)NH2 it should be R-C(=NOH)NH2 (formula stile)
Line 172 2a-h should be bold
Line 617 it should be 35 °C
Line 640 Instead of “…CHCl3-AcOEt, Organic layer…” it should be …CHCl3-AcOEt. Organic layer…
Line 1082 Instead of 37°C it should be 37 °C
Author Response
Dear Reviewer #2,
We sincerely thank you for deep analysis of our manuscript and recommendation of this article for publication in International Journal of Molecular Sciences! We revised the manuscript according to your comments:
- In the legend of Scheme 1:
- Instead of CH2Cl2 it should be formula stile CH2Cl2
- Instead of R-C(=NOH)NH2 it should be R-C(=NOH)NH2 (formula stile)
Authors: Corrected (please, see lines 107-108; marked by yellow).
- Line 172 2a-h should be bold
Authors: Corrected (please, see line 108; marked by yellow).
- Line 617 it should be 35 °C
Authors: Corrected (please, see line 638; marked by yellow).
- Line 640 Instead of “…CHCl3-AcOEt, Organic layer…” it should be …CHCl3-AcOEt. Organic layer…
Authors: Corrected (please, see line 661; marked by yellow).
- Line 1082 Instead of 37°C it should be 37 °C
Authors: Corrected (please, see line 1110; marked by yellow).

Round 2
Reviewer 1 Report
The manuscript has been correctly revised. However, there is a problem with the text in Figure 1. Please correct.
The manuscript can be accepted for publication.
Author Response
Dear Reviewer #1,
We are sincerely thankful to you for thorough analysis of our manuscript and recommendation of this article for publication in International Journal of Molecular Sciences! We revised the manuscript according to your comment:
The manuscript has been correctly revised. However, there is a problem with the text in Figure 1. Please correct.
Authors: Corrected. The text in Figure 1 was rewritten (please, see lines 93-94).
